# Technical note: Emulation of a large-eddy simulator for stratocumulus clouds in a general circulation model

Kalle Nordling[1,3], Jukka-Pekka Keskinen[1], Sami Romakkaniemi[2], Harri Kokkola[2], Petri Räisänen[1], Antti Lipponen[2], Antti-Ilari Partanen[1], Jaakko Ahola[1], Juha Tonttila[2], Muzaffer Ege Alper[1], Hannele Korhonen[1], and Tomi Raatikainen[1]

[1]Finnish Meteorological Institute, Helsinki, Finland
[2]Finnish Meteorological Institute, Kuopio, Finland
[3]CICERO Center of International Climate Research, Oslo, Norway

**Correspondence:** Kalle Nordling (kalle.nordling@fmi.fi)

**Abstract.** Here we present for the first time a proof of concept for an emulation-based method that uses a large-eddy simulations (LES) to present sub-grid cloud processes in a general circulation model (GCM). We focus on two key variables affecting the properties of shallow marine clouds: updraft velocity and precipitation formation. The LES is able to describe these processes with high resolution accounting for the realistic variability in cloud properties. We show that the selected emulation method is able to represent the LES outcome with relatively good accuracy and that the updraft velocity and precipitation emulators can be coupled with the GCM practically without increasing the computational costs. We also show that the emulators influence the climate simulated by the GCM, but do not consistently improve or worsen the agreement with observations on cloud related properties, although especially the updraft velocity at cloud base is better captured. A more quantitative evaluation of the emulator impacts against observations would, however, have required model re-tuning, which is a significant task and thus could not be included in this proof-of-concept study. All in all, the approach introduced here is a promising candidate for representing detailed cloud and aerosol related sub-grid processes in GCMs. Further development work together with increasing computing capacity can be expected to improve the accuracy and the applicability of the approach in climate simulations.

## 1 Introduction

Clouds play a major role in the Earth's climate system through their effects on the hydrological cycle and the radiative balance (Boucher et al., 2013; Rosenfeld et al., 2019). They mainly reflect the incoming short-wave solar radiation and trap outgoing long-wave radiation from the relatively warm surface. Cloud radiative effects depend on cloud macro- and microphysical properties like cloud fraction, amount of liquid water and ice, and the size of cloud particles. These properties, in turn, depend on atmospheric thermodynamics and dynamics as well as microphysical processes. Typically, cloud formation is related to updrafts driven by convection, turbulence, or larger-scale weather systems. Precipitation is one of the key cloud microphysical processes influencing cloud dynamics and lifetime.

Accurate simulation of clouds has been a long-standing challenge for general circulation models (GCMs). This is especially true of shallow clouds, in particular marine stratocumuli, which are the focus of this work (Cesana and Del Genio, 2021;

Tselioudis et al., 2021). Marine stratocumuli are highly abundant covering approximately 23 % of the ocean surface and they strongly influence the climate primarily by reflecting solar radiation (Wood, 2012; Muhlbauer et al., 2014). In spite of their

importance, GCMs still struggle to simulate marine stratocumuli realistically. In particular, for decades, most GCMs have underestimated the occurrence of stratocumulus clouds and their impacts on the radiation budget (Siebesma et al., 2004; Teixeira et al., 2011; Lauer and Hamilton, 2013; Jian et al., 2020). The difficulties in representing marine stratocumulus clouds in GCMs arise from two main reasons (e.g Kawai and Shige, 2020). On one hand, the occurrence and properties of stratocumulus clouds are determined by the interplay of a multitude of physical processes, including turbulence, convection,

cloud and precipitation microphysics and radiative transfer. On the other hand, these issues are exacerbated by the vertical resolution of GCMs that is insufficient to properly resolve boundary layer clouds and especially the inversion layer that controls them (Guo et al., 2019). The clouds are subgrid-scale features also horizontally. Individual shallow clouds have horizontal dimensions starting from a few tens of meters, while the typical grid spacing in GCMs is 50+ kilometres.

The radiative effects of marine stratocumulus are mainly determined by their cloud fraction $A$ and cloud water content $q_c$,

also known as cloud macrophysical properties. The recent review paper by Kawai and Shige (2020) separates three basic approaches to parameterize $A$ and $q_c$ in GCMs. Either $A$ is diagnostic and $q_c$ is prognostic (e.g. Sundqvist et al., 1989) or both are prognostic (e.g. Tiedtke, 1993; Wilson et al., 2008), or both are derived diagnostically based on prognostic total water content and liquid-frozen temperature variables (e.g. Smith, 1990; Watanabe et al., 2009). All these approaches require either explicit or implicit assumptions about the subgrid-scale distribution of total water content. While diagnostic cloud fraction schemes in

GCMs typically parameterize $A$ as a function of relative humidity, the parameterizations are sometimes adjusted for stratocumulus clouds by also considering the strength or presence of a temperature inversion (Slingo, 1987; Teixeira and Hogan, 2002; Stevens et al., 2013). A more ambitious approach is the use of unified parameterizations. These include the eddy-diffusivity mass flux (EDMF) approach (Siebesma et al., 2007), which seeks to unify planetary boundary layer (PBL) and shallow convective processes, and the Cloud Layers Unified by Binormals (CLUBB) scheme (Golaz et al., 2002), which attempts to unify

cloud microphysics, PBL turbulence and shallow convection schemes. Suselj et al. (2021) reported significantly improved simulation of subtropical stratocumulus clouds when using the EDMF approach in the NASA GEOS model, while Bogenschutz et al. (2013) reported more realistic simulation of the stratocumulus-to-cumulus transition in Community Atmosphere Model, version 5 (CAM5) in experiments using CLUBB. Similarly, Guo et al. (2014) reported that when using CLUBB with the GFDL atmospheric general circulation model (AM3), the simulations of coastal stratocumulus were significantly improved. Finally,

Yamaguchi et al. (2017) developed a methodology named Framework for Improvement by Vertical Enhancement (FIVE), in which selected processes relevant for the simulation of clouds are computed at higher vertical resolution. Lee et al. (2022) showed that FIVE helped to alleviate but still not eliminate the underestimation of subtropical stratocumulus clouds in the Energy Exascale Earth System Model (E3SM).

Aside from cloud macrophysics, cloud radiative effects also depend on cloud microphysical properties. One key parameter

is the cloud droplet number concentration (CDNC). CDNC depends on the available aerosol (indirect effect) and the rate of humidity increase, which is typically related to updraft velocity. The importance of aerosol and updraft velocity for droplet concentration has been widely studied. For example, Yoshioka et al. (2019) found the uncertainty in the updraft velocity to

be the second most important cause for uncertainty in aerosol radiative forcing, Sullivan et al. (2016) state that input updraft velocity fluctuations can explain as much as 61% of droplet number variability in GEOS-5, and West et al. (2014) found

$0.4 \, \mathrm{W \, m^{-2}}$ uncertainty in the aerosol indirect effect due to different estimates for updrafts in global models. Despite the known importance, most models still rely on a rather simple approach based on the resolved turbulent kinetic energy or eddy diffusivity to parameterise the vertical velocity, which is described by a single grid-scale characteristic value or by a Gaussian distribution of values representing sub-grid variability (Golaz et al., 2011; West et al., 2014; Matsui and Moteki, 2020).

Beyond the direct cloud albedo effect, cloud droplet number concentration also affects the cloud processing and wet removal

of aerosol particles and cloud water by precipitation (Matsui and Moteki, 2020). The conversion of cloud water into rain water and precipitation is based on so-called autoconversion parameterisations (e.g. Seifert and Beheng, 2001; Khairoutdinov and Kogan, 2000; Golaz et al., 2011; Kawai and Shige, 2020). Due to the simplified treatment of both the droplet size distribution and cloud dynamics, global models tend to produce warm precipitation with too little variability in strength (Jing et al., 2019). It is also common for global models that the formation of warm precipitation is too efficient (Suzuki et al., 2015). Related

to warm rain formation, attempts have been made to avoid these issues by modifying the autoconversion efficiency either by increasing the cloud droplet threshold size for precipitation formation, or simply by scaling the autoconversion strength to account for resolution differences (e.g. Golaz et al., 2011; Mülmenstädt et al., 2020). These modifications, and the mathematical dependence of autoconversion rate on CDNC contributes strongly to the spread of modelled aerosol indirect effect (Jing et al., 2019).

In an approach commonly referred to as super-parameterisation, the conventional cloud parameterisations within each climate model grid cell or vertical column are replaced with a high-resolution model. For example, cloud-resolving models (CRMs) are well-suited for describing a column of a GCM with about 1 km resolution and with accounting for additional micro-physical details (Grabowski and Smolarkiewicz, 1999; Stan et al., 2010; Khairoutdinov et al., 2005). CRMs are especially good for simulating cloud systems and they can cover multiple cloud cycles and types (Khairoutdinov and Randall,

2003). Applying the super-parameterisation approach has been shown to significantly improve e.g. the predictions of surface precipitation compared to conventional GCMs (Tao et al., 2009). However, shallow clouds are still challenging for CRMs, because the CRM resolution is insufficient for resolving even the largest scales of boundary layer turbulence, which has a dominant role for generating the updrafts in shallow clouds. Large-eddy simulators (LESs) are the best tools for such clouds, as these models account for turbulence and they have significantly higher resolution ($\lesssim$100 m) than CRMs have ($\sim$1 km). The

higher resolution allows them to be based on physics rather than parameterisations. Unfortunately, using a computationally expensive LES as a super-parameterisation is currently impossible at least for climate simulations (Grabowski, 2016; Parishani et al., 2017; Jansson et al., 2019).

Computationally efficient machine learning approaches have been used to tackle the super-parameterisation problem in climate simulations. In principle, any deterministic model such as LES can be represented by a fast statistical surrogate model

called emulator (Rasp et al., 2018; Glassmeier et al., 2019; Besombes et al., 2021; Conibear et al., 2021). Glassmeier et al. (2019) developed statistical emulators for cloud parameters derived from 159 LES runs with various inputs. Then they used the emulators as advanced interpolation tools to quantify the impacts of aerosol-cloud interactions for a wide range of cloud

conditions. Statistical emulators have been used in several occasions to replace the traditional parameterisations and sub-grid scale models in GCMs (e.g. Reichstein et al., 2019; Yuval and O'Gorman, 2020). The first applications employed neural networks trained using the regular radiation scheme to calculate the long-wave radiative budget with reduced computational cost (Cheruy et al., 1996; Chevallier et al., 1998). Similarly, an emulator based on the random forests technique has been used to replace the regular moist convection scheme of a GCM (O'Gorman and Dwyer, 2018). Machine learning approaches have also aimed at developing improved parameterisations based on training data from simulations with a higher model resolution or improved model physics. For example, Han et al. (2020), Wang et al. (2022) and Bretherton et al. (2022) used a deep learning method and training data from a super-parameterised or a kilometer-scale GCM to develop improved parameterisations for moist physics, convection and radiative fluxes. However, the super-parameterisations utilizes a kilometer-scale CRM, so the scheme is best suited for deep convective clouds. In fact, to our knowledge, there are currently no approaches that use LES and machine learning to improve moist physics for shallow clouds in a GCM.

In this study, we present a proof-of-concept for using LES-based emulators to describe processes driving marine stratocumulus cloud properties and life cycle in the global aerosol-chemistry-climate model ECHAM. Our focus is on updraft velocity and warm rain formation which are among the main sources of uncertainty in the aerosol radiative forcing estimates of current climate models (Donner et al., 2016; Jing et al., 2019; Bougiatioti et al., 2020; Yoshioka et al., 2019). These processes were chosen because, on the one hand, updraft velocity is the dominant factor in hydrometeor number variability (Sullivan et al., 2016) and its emulation is technically relatively straightforward (Ahola et al., 2022). On the other hand, current schemes of precipitation formation in GCMs are quite simplified producing known biases (Jing et al., 2019; Suzuki et al., 2015). Emulating precipitation formation allows for increased realism for this highly important process, for example, by accounting for the impacts of sub-grid variability in cloud properties. Finally, an important practical aspect is that emulators for updraft speed and rain water formation can be used in ECHAM without major structural changes in the model parameterisations.

Below we present the emulator development work and compare the obtained results with those from the default version of the same GCM as well as against observations. Our aim is to demonstrate that the new approach results in stable, physically sound GCM simulations, which has been an issue with some previous methods (Yuval and O'Gorman, 2020; Yuval et al., 2021; Brenowitz et al., 2020). We also discuss the lessons learned during the emulator development and present ideas for the way forward.

## 2 Methods

### 2.1 Model description

#### 2.1.1 UCLALES-SALSA

The LES model used in this study is UCLALES-SALSA (Tonttila et al., 2017). The model includes both the detailed sectional aerosol and cloud microphysics module SALSA (Kokkola et al., 2018) and the double-moment bulk microphysics parameterisation (Seifert and Beheng, 2001; Stevens and Seifert, 2008) used in the original UCLALES (Stevens et al., 1999, 2005).

Our previous updraft velocity emulator development work (Ahola et al., 2022) showed that the high computational cost of the SALSA microphysics limited the number of emulator training simulations to a level that was not adequate for practical applications. Therefore, we use simulations from (Ahola et al., 2022) made with the double-moment bulk microphysics. In this model version, cloud water mixing ratio is diagnosed using the saturation adjustment method. Cloud droplet number concentration is given as an input parameter and is assumed to be the same for all cloudy grid cells. The double-moment warm-rain scheme

includes parameterisations for autoconversion, rain water evaporation, accretion, and sedimentation (Seifert and Beheng, 2001; Stevens and Seifert, 2008). Microphysics is coupled with the LES, which simulates the turbulent atmospheric flows, so that they interact via latent heating. The partitioning of water also influences buoyancy and radiative heating and cooling, which are important drivers of updraft velocity.

### 2.1.2 ECHAM

The GCM simulations were carried out with the ECHAM (ECHAM6.3-HAM2.3-MOZ1.0) global aerosol-chemistry-climate model (Schultz et al., 2018). It consists of the atmospheric model ECHAM (Stevens et al., 2013), aerosol model HAM (Kokkola et al., 2018; Tegen et al., 2019), and chemistry model MOZ (Schultz et al., 2018). The aerosol model HAM includes two configurations for the aerosol size distribution: modal treatment M7 (Tegen et al., 2019) and the sectional scheme SALSA (Kokkola et al., 2018). From here on, we refer to these model setups as ECHAM-M7 and ECHAM-SALSA. We used the

standard ECHAM-M7 setup to generate the input data for the LES training simulations (see Sect. 2.2). However, because our original aim was to include the aerosol impacts by using SALSA microphysics in the LES runs (i.e., UCLALES-SALSA), the emulators were implemented into ECHAM-SALSA. Hence, in the following we focus on ECHAM-SALSA.

Aerosol-cloud interactions in the model are simulated using the two-moment cloud microphysics scheme of Lohmann (2008) and Lohmann and Hoose (2009) with modifications described by Lohmann and Neubauer (2018). In addition, SALSA uses

the sectional aerosol size distribution scheme of Abdul-Razzak and Ghan (2002) instead of the modal setup used in Lohmann and Neubauer (2018). In the default setup of ECHAM, a lower bound of $40\,\mathrm{cm}^{-3}$ for CDNC has been set (Lohmann et al., 1999). This is a common practice in global models to avoid CDNC values which are considered too low (Hoose et al., 2009). However, in ECHAM, this reduces the sensitivity of CDNC to changes in updraft velocity. This is why we reduced the lower bound to the value of $10\,\mathrm{cm}^{-3}$. This allowed for better assessing the impact of the updraft velocity on CDNC but on the other

hand resulted in a high aerosol radiative forcing (see Sect. 3.3.4).

Cloud fraction is parameterised as a function of relative humidity (RH), using the assumed humidity distribution function scheme developed by Sundqvist et al. (1989). As detailed in Sect. 3.2.3 in Stevens et al. (2013), the critical relative humidity for cloud formation depends on pressure, and in order to enhance the simulation of stratocumulus clouds, a 100% cloud fraction is assumed already at RH=90% if there is a temperature inversion below $700\,\mathrm{hPa}$. Mixing ratios of cloud liquid water and cloud

ice are treated prognostically. The prognostic equations follow the approach described by Lohmann and Roeckner (1996), which accounts for the transport by the adiabatic circulations, exchange terms that convert water from one of the prognostic phases (vapor,liquid, solid) to another, and conversion to large-scale precipitation.

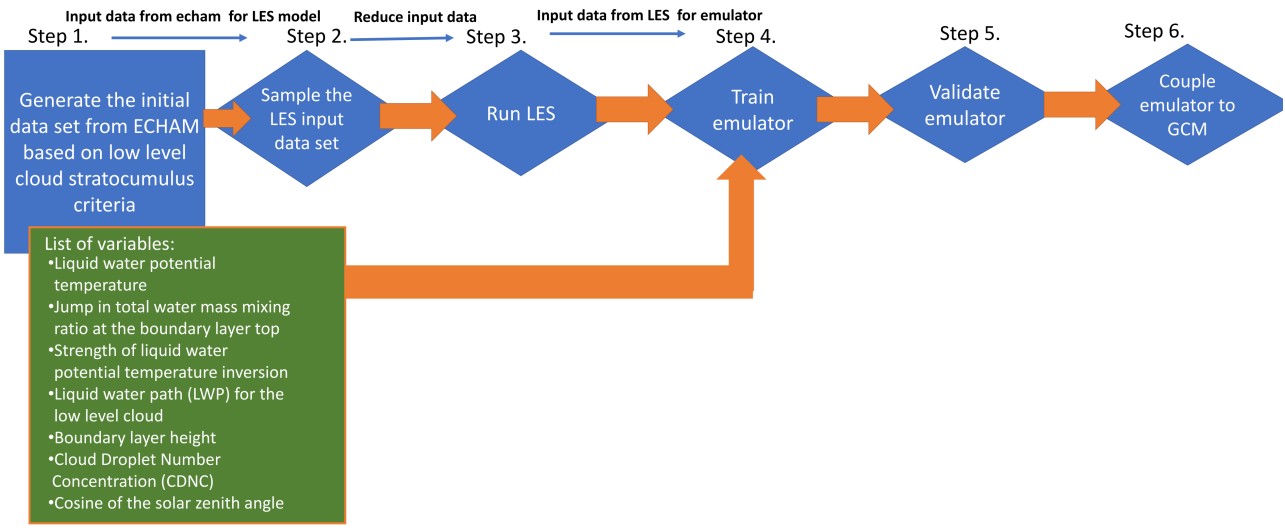

**Figure 1.** Flowchart for developing and coupling the emulator to ECHAM GCM.

Vertical velocity in the default version of ECHAM is based on a single characteristic vertical velocity, which is calculated as the sum of large-scale and turbulent kinetic energy-dependent velocity terms (Lohmann et al., 2007). In the two-moment microphysical module of ECHAM, the rain water formation rate is calculated using the autoconversion parameterisation of Khairoutdinov and Kogan (2000) for each model level. The rate is proportional to the ratio of cloud liquid water content to cloud droplet number concentration.

ECHAM uses a spectral representation of atmospheric dynamics along with a flux-form semi-Lagrangian scheme for tracer transport. In our model configuration, the horizontal resolution was T63 which corresponds to approximately $1.9° \times 1.9°$. In the vertical, the model had 47 layers with a terrain-following hybrid vertical coordinate representation (Stevens et al., 2013) with the model top at $0.01\,\mathrm{hPa}$. Sea surface temperatures (SSTs) and sea ice cover were fixed to the monthly mean climatologies provided by the Atmospheric Model Intercomparison Project (AMIP). For simulations with warmer climate, SSTs were increased uniformly by $4\,\mathrm{K}$. Simulations were run for 13 years (1997-2009) and the last 10 years were used in the analysis.

## 2.2 Emulator development

The main steps of the emulator development are presented in the flowchart in Fig. 1. Steps 1–3, which include the LES and the initial ECHAM-M7 simulations for the selected emulation approach are obtained from Ahola et al. (2022), so only a brief overview is given here. For this study, we selected the Gaussian process emulation (GPE: O'Hagan (1978) and O'Hagan (2006)) technique, which is suitable for representing the computationally expensive LES. Based on the previous LES runs, we developed emulators for cloud base updraft velocity and rain water formation rate. These new emulators were validated using methods similar to those in Ahola et al. (2022).

The background information for the emulator training data set was obtained from a year-long ECHAM-M7 simulation. The first step in emulator development was an initial analysis of the GCM (ECHAM-M7), where the conditions for using the emulator and the emulator variables were determined (Ahola et al., 2022). The conditions for using the emulator limited the approach to shallow marine clouds. In practice, this means GCM columns with a warm cloud (at least 90% liquid) above open sea surface (no fog) and below the 700 hPa vertical level (a low cloud). Columns with significant mid or high-level clouds (more than half of the condensed water located above the low cloud) were excluded as these would have had an impact on the radiative fluxes incident at the top of the low cloud. These conditions for identifying low clouds are fairly relaxed, at least from the meteorological point of view, but they allow applying the emulators to a reasonably high fraction of ECHAM columns, also beyond archetypal stratocumulus regions. It should be pointed out, however, that the emulator is effectively limited to stratus and stratocumulus clouds, as the LES simulations were initialized with a cloud fraction of unity. The reason for this is that the simulation area ($10 \times 10$ km) and the length of the LES simulations (3.5 h) are insufficient to represent properly mesoscale simulations associated with shallow cumulus clouds (Saffin et al., 2023). Producing an emulator that better represents also shallow cumulus clouds would be an important goal for future work, but it is currently beyond our computational resources.

The emulator input variables are a set of six or seven scalars describing the identified shallow marine cloud columns in a simplified way (Ahola et al., 2022). The additional seventh input variable for the daytime simulations is the (cosine of) solar zenith angle. The variables describing the cloud were derived subject to the assumption of a well-mixed cloud-topped boundary layer, with a constant total water mass mixing ratio and liquid water potential temperature from the sea surface up to the top of the boundary layer. The inversion layer above the boundary layer top is described by the corresponding humidity and temperature jumps. In order to ensure accurate cloud water content, the humidity variable for the emulator is liquid water path (LWP), but with the above-mentioned assumption it can be related to a certain total water mass mixing ratio. The boundary layer is thus described with three variables: the cloud top height, LWP of the low cloud and liquid water potential temperature, which is the minimum value in an ECHAM column within the distance of two grid cells from the boundary layer cloud. The humidity and temperature jumps are calculated as the difference between the maximum and minimum values of total water mass mixing ratio and liquid water potential temperature within the distance of two grid cells from the cloud. The sixth variable is CDNC, which is averaged over cloudy grid cells.

As mentioned above, the initial data set of the emulator input variables was obtained from a year-long ECHAM-M7 simulation with a one-year spin-up. The conditions above resulted in almost six million ECHAM columns, which are described by the six or seven variables, so the second step in emulator development was to sample a representative subset for the LES runs. For this Ahola et al. (2022) applied the binary space partitioning (BSP; Fuchs et al. (1980)) approach. The BSP method uses variable distributions to construct the emulator training datasets so that they well represent the variable space and therefore is beneficial for the emulator. Because nighttime simulations are independent of the solar zenith angle, separate day and night samples were collected. Both of these samples contain 500 cloud cases.

The third step in the emulator development was running the LES simulations. The five meteorological input variables describing the well-mixed cloud-topped boundary layer were used to reconstruct the initial temperature and humidity profiles for the LES runs, again assuming a well-mixed cloud-topped boundary layer. CDNC and solar zenith angle were inputs for the

cloud microphysics and daytime radiative transfer calculations, respectively. The other model settings and the simulations are described in Ahola et al. (2022).

Here we used the LES data set from Ahola et al. (2022) for calculating cloud base updraft velocities and rain water formation rates for the emulator training (step 4). The updraft velocity was calculated as the average of positive (updraft) cloud base vertical velocities from the last hour of each 3.5 h simulation (Romakkaniemi et al., 2009). Three daytime simulations had no clouds during the last hour, so these cases were excluded from the emulator training. Rain water formation rate is based on domain mean vertical integrals of the removal (below-cloud evaporation and surface precipitation) rates that are multiplied by minus one. We used the removal rates instead of the actual formation (autoconversion and accretion) rates mainly to avoid the impact of spin-up (the first 1.5 h) on the parameterized autoconversion. Namely, autoconversion has the highest rates when the process is switched on after the spin-up. Typically, the rates decreased and reached a steady state within 30 min, but there were a few exceptions with high rates until the end of simulation. Precipitation and below-cloud evaporation depend on the rain droplet size distribution, and its development is limited by the accretion process, so unrealistic rates are much less frequent. Precipitation rates will eventually decrease due to the removal of condensable water. For this reason, rain water formation rate was calculated as the average of the three largest values in each LES simulation representing a developed rain drop size distribution.

Separate emulators were trained for each output (updraft velocity and rain water formation rate) and for daytime and night-time because the daytime emulators have solar zenith angle as an additional input. In all emulator applications mentioned here we used the GPF Fortran library (https://github.com/ots22/gpf) extended for our purpose. In practice, the emulator training means optimizing hyper-parameters of the covariance function (here we added a new covariance function combining squared exponential and linear terms) so that the emulator predictions match with the target outputs (Rasmussen and Williams, 2005). Emulator predictions are based on the hyper-parameters and covariance matrices calculated from the training data set (the LES inputs and the corresponding outputs). We used an offline emulator training approach where these emulator parameters were first saved to a data file and later the parameters were simply read to the emulator coupled to the GCM (step 6). Practical details about the coupling are given in Sect. 2.3.

To evaluate the emulators' accuracies, we used the leave-one-out cross-validation method (step 5). In this method, one member of the training data set is left out of the training, and a validation emulator is trained using the remaining data. The trained validation emulator is then used to predict the output of the member not used for the training. This process is repeated for each member of the training data set. With this approach, the validation data is independent of the training data and corresponds to emulators trained with as complete training data as possible. A large training data set is important in the validation as the emulator performance improves with more training data samples added. Furthermore, in leave-one-out cross-validation, the training data sets of the validation models differ only by one data sample of the training data used to train the final emulator. Therefore, the leave-one-out cross-validation method gives a realistic understanding of the final emulator's accuracy. Emulator validation results are shown in Sect. 3.1.

## 2.3 Emulator-ECHAM coupling

We have implemented the four different emulators (updraft velocity and rain water production rate for day and night) to ECHAM-SALSA. All following simulations were made using this model setup, so from now on we drop the suffix SALSA. In addition, we use the term precipitation instead of rain water production rate, because precipitation in ECHAM is directly related to the rain water production rate. The daytime versions of the emulators are activated when the cosine of the zenith angle is positive. Otherwise the nighttime versions are used. In all cases, emulation is applied only in columns where the column selection criteria, as defined earlier in Sect. 2.2, are met.

In the default version of ECHAM, the updraft velocity is calculated as the sum of large-scale and turbulent vertical velocity. It is then used in the cloud droplet activation scheme (Abdul-Razzak et al., 1998; Abdul-Razzak and Ghan, 2002). The coupling of the updraft velocity emulator to ECHAM was straightforward: the original updraft velocities within the cloud layer were simply replaced with the value given by the emulator.

The inclusion of the precipitation emulator was more complicated. As clouds can span several model layers in ECHAM but the emulator provides only a single vertically integrated value for each column, an approach for the distribution of emulated precipitation to model levels was required. In the two-moment microphysical module of ECHAM, the autoconversion rate is calculated using the approach of Khairoutdinov and Kogan (2000) for each model level, which in turn modifies the simulated cloud water mass mixing ratio. In an attempt to preserve the vertical structure of precipitation, we chose to divide the emulated column precipitation to cloudy levels in the same manner. More specifically, the emulated precipitation $P_e$ ($\mathrm{kg\,m^{-2}\,s^{-1}}$) for layer $i$ is calculated as

$$P_{e,i} = \frac{P_e q_i^{2.47} N_i^{-1.79}}{\sum_j q_j^{2.47} N_j^{-1.79}}, \tag{1}$$

where $q$ is the grid box mean cloud liquid water content ($\mathrm{kg\,kg^{-1}}$) and $N$ is the cloud droplet number concentration ($\mathrm{m^{-3}}$) and index $j$ covers the low cloud. The emulated precipitation is then enforced on each cloudy layer by scaling the terms that make up the precipitation in ECHAM (i.e. autoconversion with droplets inside the layer, and accretion with rain drops from above layers). To ensure conservation of water, the maximum amount of precipitation after the application of the precipitation emulator is limited by the available cloud liquid water within the grid box. Here the term cloudy refers to grid boxes in which the liquid water content is at least $0.01\,\mathrm{g\,kg^{-1}}$.

## 2.4 Experiment design

The emulators implemented to ECHAM were tested in a set of simulations, the results of which were compared against the standard ECHAM set-up as well as against observations listed in Sect. 2.5. The eight ECHAM simulations are summarized in Table 1. They consist of four present day simulations (CTRL, EMU-PR, EMU-UP and EMU-BOTH), two simulations with pre-industrial aerosol emissions (PI-CTRL and PI-EMU-BOTH), and two simulations with warmer climate (WARM-CTRL and WARM-EMU-BOTH). In the control (CTRL) simulation, the emulators are invoked only for diagnostic purposes, and default parameterisations are used for integrating the model state forward in time. In EMU-BOTH updraft and precipitation emulators

**Table 1.** ECHAM simulations conducted.

| Simulation | Aerosol | Emulator(s) |
|---|---|---|
| CTRL | Present day | None |
| EMU-PR | Present day | Precipitation only |
| EMU-UP | Present day | Updraft only |
| EMU-BOTH | Present day | Precipitation and updraft |
| PI-CTRL | Pre-industrial | None |
| PI-EMU-BOTH | Pre-industrial | Precipitation and updraft |
| WARM-CTRL | Present day | None |
| WARM-EMU-BOTH | Present day | Precipitation and updraft |

are both applied, whereas in EMU-UP only the updraft emulators and in EMU-PR only the precipitation emulators are active. For estimating the aerosol effective radiative forcing (ERF), we conducted the PI-CTRL and PI-EMU-BOTH runs, which are identical to the CTRL and EMU-BOTH runs except that aerosol emissions are from year 1850. ERF is calculated as the difference between the net top-of-atmosphere radiative fluxes in the present-day and pre-industrial runs. The two simulations with warmer climate were made for exploring how the emulators perform in climatic conditions different from the training dataset, and how they impact the climate sensitivity of the ECHAM model. For this we follow the method by Cess et al. (1990) where the warm climate is produced by increasing the SST uniformly by +4 K.

For the simulations with present-day aerosol emissions, we used the ACCMIP emission data (Lamarque et al., 2010) until the end of year 2004 and after that the emissions come from the representative concentration pathway (RCP) projection RCP4.5 (Van Vuuren et al., 2011). For the pre-industrial simulations, we used the ACCMIP aerosol emission data for the year 1850. Dust, sea salt, and marine dimethylsulfide (DMS) emissions were calculated on-line based on 10-meter wind speed. Dust emissions are based on Tegen et al. (2002) with modifications described by Cheng et al. (2008) and Heinold et al. (2016), sea salt emissions are based on Guelle et al. (2001), and online DMS emissions are according to Kloster et al. (2006).

## 2.5 Observations

The present-day simulations with and without emulators are compared against observational data sets of surface precipitation and shortwave cloud radiative effect (SW-CRE). The precipitation data set comes from the Global Precipitation Climatology Project (GPCP) (Adler et al., 2012) and for SW-CRE we use the Clouds and the Earth's Radiant Energy System (CERES) Energy Balanced and Filled (EBAF) top-of-atmosphere (TOA) edition-4.0 data product (Loeb et al., 2018). In addition, we use cloud cover data from Stubenrauch et al. (2013) and LWP data from Multi-Sensor Advanced Climatology of LWP (Elsaesser et al., 2016). These same observational data sets were used by Neubauer et al. (2019) in their evaluation of the ECHAM6.3-HAM2.3 model. For the comparison, we averaged the model and observational data from year 2000 to 2009.

## 3 Results

First we show a brief validation of the emulators by comparing the emulator, LES and ECHAM predictions for updraft and
precipitation. Then we show where and how frequently the emulators are used in ECHAM and how this influences the simulated cloud states. Finally, we show how the emulators influence the simulated climate by examining cloud cover, surface precipitation, shortwave cloud radiative effect, and aerosol effective radiative forcing.

### 3.1 Emulator evaluation

Figure 2 shows the correlation between stand-alone emulator predictions and LES outputs (i.e., the truth) for the daytime and
nighttime rain water production rates and updraft velocities. Here the emulator predictions are based on the leave-one-out cross-validation method (see Sect. 2.2). The insets show the corresponding error distributions as histograms. For clarity, the histograms cover a limited range of values, and those exceeding the lower or upper limits are added to the first or last bin, respectively.

Figure 2a shows that most rain water production rates are close to zero and the larger rates cover a wide range of values
from less than $0.1 \, \mathrm{kg \, m^{-2} \, day^{-1}}$ to well above $10 \, \mathrm{kg \, m^{-2} \, day^{-1}}$. The wide range of possible values can explain the few clear outliers (differences exceeding $5 \, \mathrm{kg \, m^{-2} \, day^{-1}}$), but otherwise the emulator is able to reproduce the LES predictions relatively well (95% of the absolute errors are less than $0.33 \, \mathrm{kg \, m^{-2} \, day^{-1}}$). Pearson's correlation coefficient, mean error (bias), mean absolute error (MAE) and root mean square error (RMSE) are 0.925, $-0.0084 \, \mathrm{kg \, m^{-2} \, day^{-1}}$, $0.108 \, \mathrm{kg \, m^{-2} \, day^{-1}}$ and 0.526 $\mathrm{kg \, m^{-2} \, day^{-1}}$, respectively. Figure 2b shows that updraft velocities and their errors are more evenly distributed. Although
there are a few outliers where the difference exceeds $0.2 \, \mathrm{m \, s^{-1}}$, 95% of the absolute errors are still less than $0.1 \, \mathrm{m \, s^{-1}}$. Pearson's correlation coefficient, bias, MAE and RMSE are 0.914, $0.0011 \, \mathrm{m \, s^{-1}}$, $0.033 \, \mathrm{m \, s^{-1}}$ and $0.047 \, \mathrm{m \, s^{-1}}$, respectively.

Figure 3 shows rain water production rate and updraft velocity distributions from the LES, ECHAM and emulators. The four different cases are:

- LES: the original emulator training data calculated from the LES outputs.

- LES emulator: stand-alone emulator predictions from the leave-one-out cross-validation shown in Fig. 2.

- ECHAM: outputs from the default parameterisations collected from the first year (2000) of the control simulation from columns that meet the column selection criteria.

- ECHAM emulator: outputs from the implemented emulators collected from diagnostic emulator calls for the first year (2000) of the control simulation from columns that meet the column selection criteria.

The updraft velocity distributions from LES and the LES emulators are very similar (Fig. 3b), which means that the emulator reproduces the distribution accurately although individual predictions can be noisy (Fig. 2b). Because rain water production rates cover several orders of magnitude, the distributions are shown in the logarithmic scale (Fig. 3a). The LES predicts a positive rain water production rate for any cloud, which explains the high occurrence of practically negligible values lower

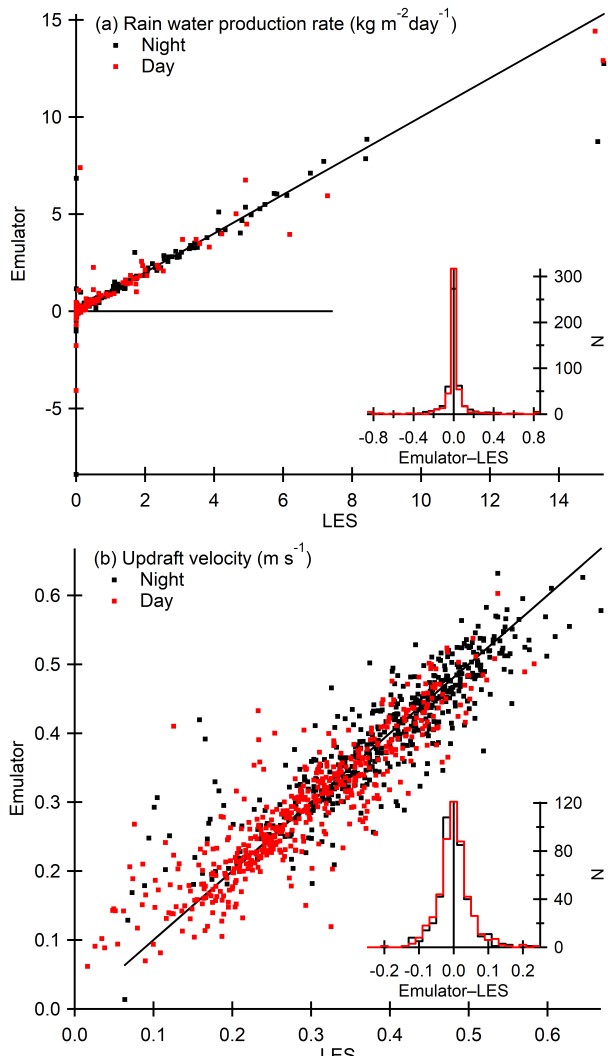

**Figure 2.** Correlation between stand-alone emulator predictions and LES outputs for nighttime (black markers) and daytime (red markers) rain water production rates (panel a) and updraft velocities (panel b). The insets show the corresponding error distributions as histograms.

than $10^{-3}$ kg m$^{-2}$ day$^{-1}$. Because the emulator predictions include a small noise term, absolute values smaller than $10^{-3}$
kg m$^{-2}$ day$^{-1}$ are being cut out while the values near $10^{-2}$ kg m$^{-2}$ day$^{-1}$ have a higher probability. Because of the noise, the emulator occasionally predicts negative values (see Fig. 2). Negative values are not shown in the logarithmic distribution, but these explain the missing probability density (area below each line). The LES and the emulator distributions agree quite well above $10^{-2}$ kg m$^{-2}$ day$^{-1}$ where the noise term becomes insignificant.

Comparison between LES and ECHAM shows that the rain water production rates from ECHAM are generally higher than
those from the LES (Fig. 3a). On the other hand, the default ECHAM values and the ECHAM emulator-based rates have similar

magnitudes but the emulator-based rates have lower frequencies due to the missing values smaller than $10^{-3} \, \mathrm{kg \, m^{-2} \, day^{-1}}$. Overall, the lower likelihood means that the rain water production emulator reduces precipitation. This is most likely due to the fact that the idealized cloud-topped marine boundary layer initial setting for the LES simulations favors drizzle if any precipitation at all. In contrast, the ECHAM data can include some less-ideal and more heavily precipitating cases (e.g., cases influenced by synoptic-scale weather systems).

The distributions of updraft velocities from the LES (and the emulators) differ from those from ECHAM, which are dominated by small values but have occasionally extremely high values exceeding $10 \, \mathrm{m \, s^{-1}}$ (Fig. 3b). Due to these high values the average vertical velocity is $0.6 \, \mathrm{m \, s^{-1}}$. Both the high and low values are missing from the LES distributions. The LES and the emulators show that daytime and nighttime updraft distributions differ, which is not seen in ECHAM. Lower daytime updrafts can be expected, because solar radiation reduces cloud-top radiative cooling which is one of the main mechanism generating turbulence in marine environments (e.g., Lilly, 1968). Overall, the updraft velocity emulators are able to produce reasonable updraft velocity distributions and also account for the differences between day and night (Zheng et al., 2016), thus improving the realism of the ECHAM simulation in this respect.

### 3.2 Cloud regions for the emulators

Figure 4 depicts how often the emulators were applied or could have been applied in the different simulations, i.e. how often the criteria outlined in Sect. 2.2 are met in each model run. Although the emulation criteria are not the same as those used to identify stratocumulus clouds, the frequency of emulator calls (18.8-21.2 %) corresponds well with the estimated stratocumulus coverage of 23 % over oceans (Wood, 2012). The emulation criteria are met most frequently at midlatitudes and in the marine regions west from North and South America and Africa, which are known for persistent stratocumulus cloud decks (Struthers et al., 2013; Neubauer et al., 2014). We will be referring to these three specific regions as Californian, Peruvian and Namibian stratocumulus regions. The geographical extent of these areas is depicted in Fig. 4 based on the boundaries defined by Partanen et al. (2012). Over these three regions the emulators were applied more than 40 % of the time.

The frequency at which the emulators are used (or could be used, for CTRL) differs between the four simulations. This means that the use of the emulators impacts atmospheric conditions and cloud properties, and hence affects when the criteria for calling the emulators are met in the subsequent time steps. Overall, using any emulator increases the frequency from 18.8 % of the CTRL simulation. In the simulation EMU-PR (Fig. 4c), the emulation criteria are met on average 19.8 % of the time in marine columns, and the number increases to 20.3 % in EMU-UP (Fig. 4d). It is also consistent that applying both emulators simultaneously (simulation EMU-BOTH, Fig. 4b) leads to a value of 21.2 % which is higher than those from the separate emulators.

### 3.3 The impact of the emulators on climate simulations

In this section, the impacts of the emulators on the climate simulated by ECHAM are reported. Specifically, we examine how the different emulator combinations affect clouds, surface precipitation, cloud shortwave radiative effects, and aerosol effective radiative forcing estimates. Section 3.3.5 shows how the emulators affect the climate sensitivity of ECHAM.

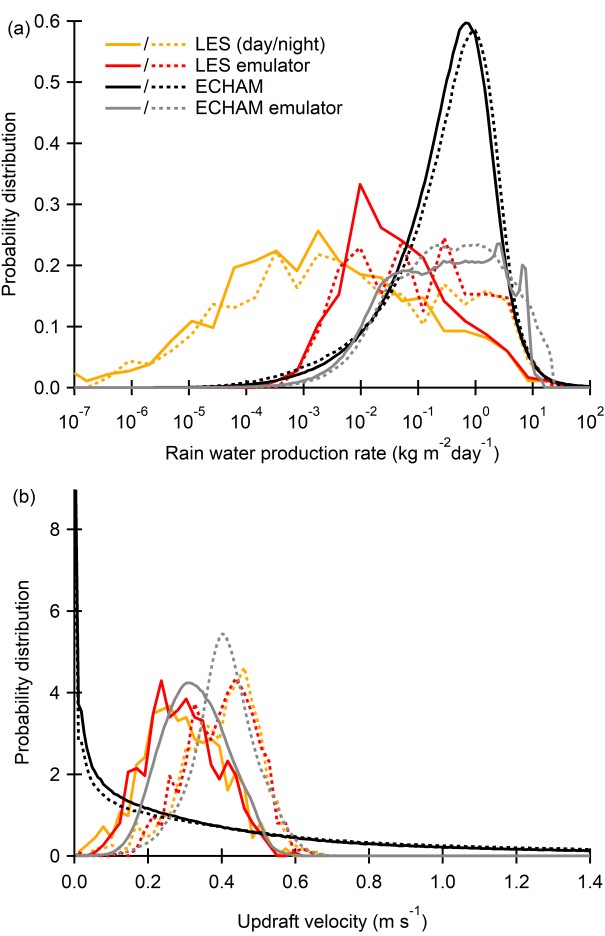

**Figure 3.** Distributions of rain water production rates (panel a) and updraft velocities (panel b) from the LES (orange), the LES emulator (red), ECHAM (black) and the emulators implemented in ECHAM (gray). Solid and dashed lines are for daytime and nighttime distributions, respectively.

### 3.3.1 Clouds

The emulators have direct and indirect impacts on various cloud variables, and here we show their impacts on mean cloud cover (Fig. 5 and Table 2) and LWP (Table 3). The tables show area-weighted mean values from simulations and observations for the three stratocumulus regions shown in Fig. 4 and the global mean. ECHAM cloud cover is evaluated using the maximum-random overlap assumption, without applying a satellite simulator. It is acknowledged that this brings some uncertainty to the comparison with satellite data.

Figure 5 and Table 2 show that while the control simulation captures the observed global annual mean cloud cover well (with a bias of 0.01), the regional biases can be larger. It is evident that the emulators have only a small impact on cloud cover and

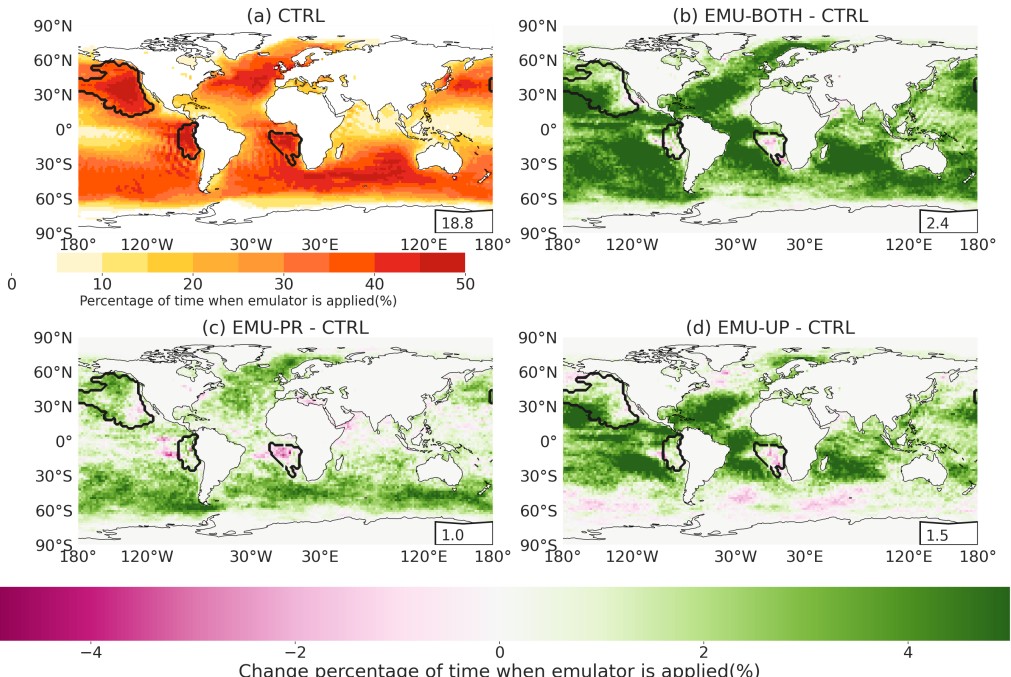

**Figure 4.** Percentage of model time steps when the emulator could have been or has been called in each marine column in simulation a) CTRL, b) EMU-BOTH, c) EMU-PR, and d) EMU-UP. Panel a shows the absolute values for CTRL while panels b–d show the difference from the CTRL (note the different color scales). The insets show the mean percentages over the marine columns. The boundaries of the Californian, Peruvian and Namibian stratocumulus regions as defined by Partanen et al. (2012) are shown with thick lines.

**Table 2.** Mean cloud cover for the three stratocumulus regions and globally from the three different emulator configurations, control run and from observations.

|          | Californian | Peruvian | Namibian | Global |
|----------|-------------|----------|----------|--------|
| CTRL     | 0.65        | 0.65     | 0.62     | 0.69   |
| EMU-BOTH | 0.66        | 0.65     | 0.62     | 0.69   |
| EMU-PR   | 0.65        | 0.65     | 0.61     | 0.69   |
| EMU-UP   | 0.65        | 0.64     | 0.61     | 0.69   |
| OBS      | 0.78        | 0.72     | 0.60     | 0.67   |

that they do not consistently improve or worsen the agreement with observations. This is also seen in Table 4 where correlation coefficients and root mean square errors between observations and different simulations are almost the same.

Table 3 shows that in the EMU-BOTH and EMU-PR experiments, cloud LWP is significantly larger than in the CTRL experiment, while EMU-UP is close to CTRL. This can be explained based on Fig. 3, which shows that the precipitation emulator produces less rain water than the original ECHAM parameterisation. Since the long-term and large-scale averages

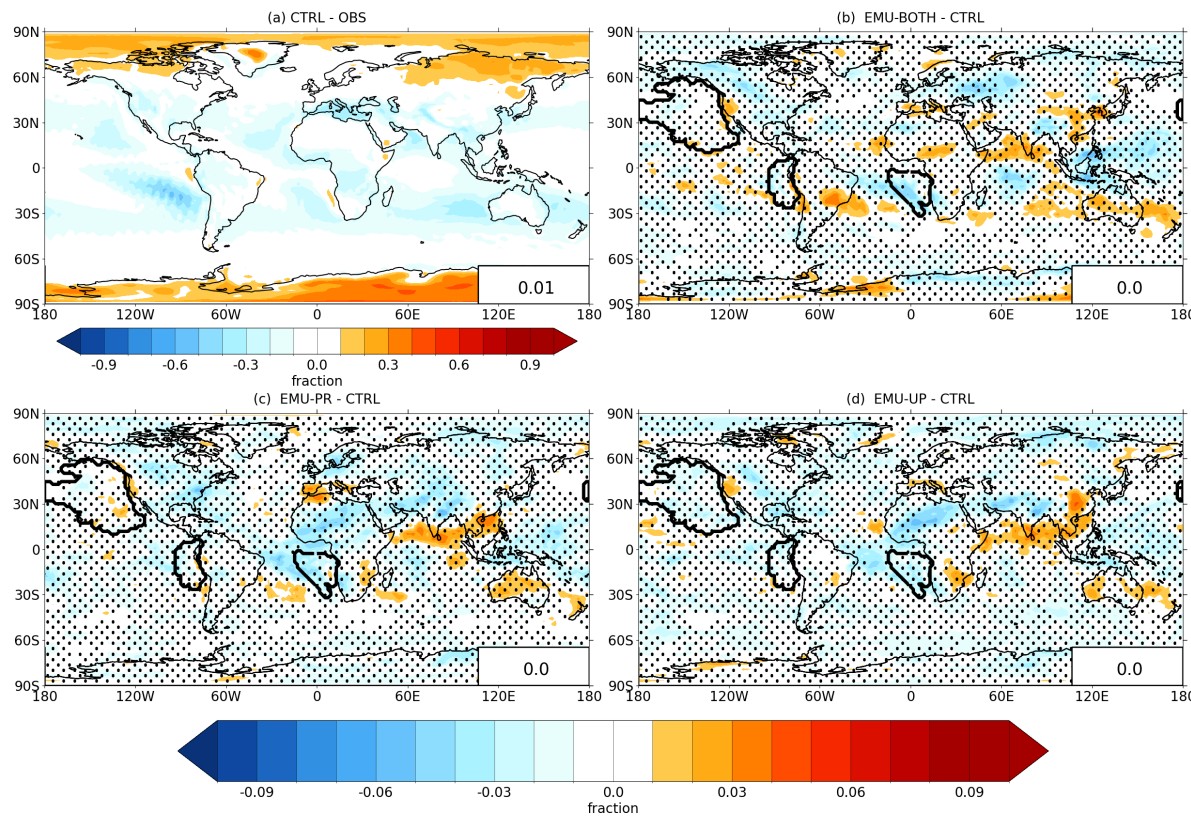

**Figure 5.** Comparison of annual mean cloud fraction between a) the CTRL simulation and observations, b) CTRL and EMU-BOTH, c) CTRL and EMU-PR, and d) CTRL and EMU-UP. Dots indicate areas where the difference is not statistically significant at $p > 0.05$. Identified marine stratocumulus regions are shown in each panel with thick black line. Global mean values are shown in the lower right corners.

of precipitation are largely controlled by surface evaporation, which changes only slightly between the different ECHAM experiments, it follows that also precipitation can only change slightly. Therefore, when using the precipitation emulator, a larger cloud liquid water amount is maintained and it generates a precipitation amount close to that in CTRL. It is also seen

from Table 3 that the LWP values for EMU-BOTH and EMU-PR clearly exceed the observational estimate for the three stratocumulus regions.

Some further insight on the impact of the emulator on cloud-related quantities is obtained by considering their vertical profiles. Fig. 6 shows the profiles of cloud water mixing ratio, CDNC, updraft velocity and cloud fraction for the three stratocumulus regions from the four ECHAM simulations. Compared to CTRL, lower maximum updraft peaks can be seen for

EMU-UP and EMU-BOTH for the clouds below 850 hPa in column 3. This is related to the distribution of the emulator-based updraft velocities, which have lower mean values than those from ECHAM (Fig. 3). The highest updraft velocities below 850 hPa thus decrease when using the updraft emulator, and this leads to decreased CDNC in these model layers. The updraft

**Table 3.** Mean LWP $(\mathrm{g\,m^{-2}})$ for the three stratocumulus regions and globally from the three different emulator configurations, control run and from observations.

|  | Californian | Peruvian | Namibian | Global |
|---|---|---|---|---|
| CTRL | 103 | 89 | 57 | 78 |
| EMU-BOTH | 117 | 96 | 64 | 86 |
| EMU-PR | 114 | 96 | 64 | 84 |
| EMU-UP | 106 | 89 | 58 | 81 |
| OBS | 86 | 71 | 44 | 80 |

**Table 4.** Correlation coefficient (r) and root mean square error (RMSE) between observations and the control run and the three different emulator configurations.

|  | CTRL | | EMU-BOTH | | EMU-UP | | EMU-PR | |
|---|---|---|---|---|---|---|---|---|
|  | r | RMSE | r | RMSE | r | RMSE | r | RMSE |
| Cloud fraction (-) | 0.73 | 0.17 | 0.74 | 0.17 | 0.74 | 0.17 | 0.74 | 0.17 |
| Precipitation $(\mathrm{mm\,day^{-1}})$ | 0.82 | 1.18 | 0.83 | 1.16 | 0.83 | 1.15 | 0.82 | 1.17 |
| SW-CRE $(\mathrm{W\,m^{-2}})$ | 0.85 | 11.33 | 0.84 | 12.39 | 0.84 | 11.89 | 0.85 | 11.87 |

emulator has an opposite effect at altitudes above 850 hPa. However, for clouds at these altitudes, the emulator is used less frequently than for clouds at lower levels, and therefore, emulator impacts on the average updraft velocity become less clear. For this reason all ECHAM experiments have similar rather small updraft velocities above 850 hPa and also the CDNC values become more similar. Consistent with the LWP values discussed above, the use of the precipitation emulator results in clearly higher cloud water content, especially below 850 hPa. While the clouds in the Peruvian and Namibian stratocumulus regions occur mostly below 850 hPa, clouds in the Californian stratocumulus region reach altitudes above 850 hPa, which explains the relatively high precipitation rates examined in the next section. Overall, the emulators have a negligible effect on cloud cover as seen in panels d, h and l.

### 3.3.2 Precipitation

Figure 7a illustrates the bias in the CTRL simulation compared to the observed surface precipitation. The CTRL experiment overestimates precipitation over large areas of the tropical oceans, especially in the Indian and Pacific oceans. The global average bias is $0.29\,\mathrm{mm\,day^{-1}}$. Panels b–d show the impacts of the emulators on precipitation simulated by ECHAM (note the different color scale). Globally, the impact of the emulators is small as most of the time the emulator is not applied. The local precipitation biases are reduced slightly in some regions but increased in others. Table 4 shows correlation coefficients between

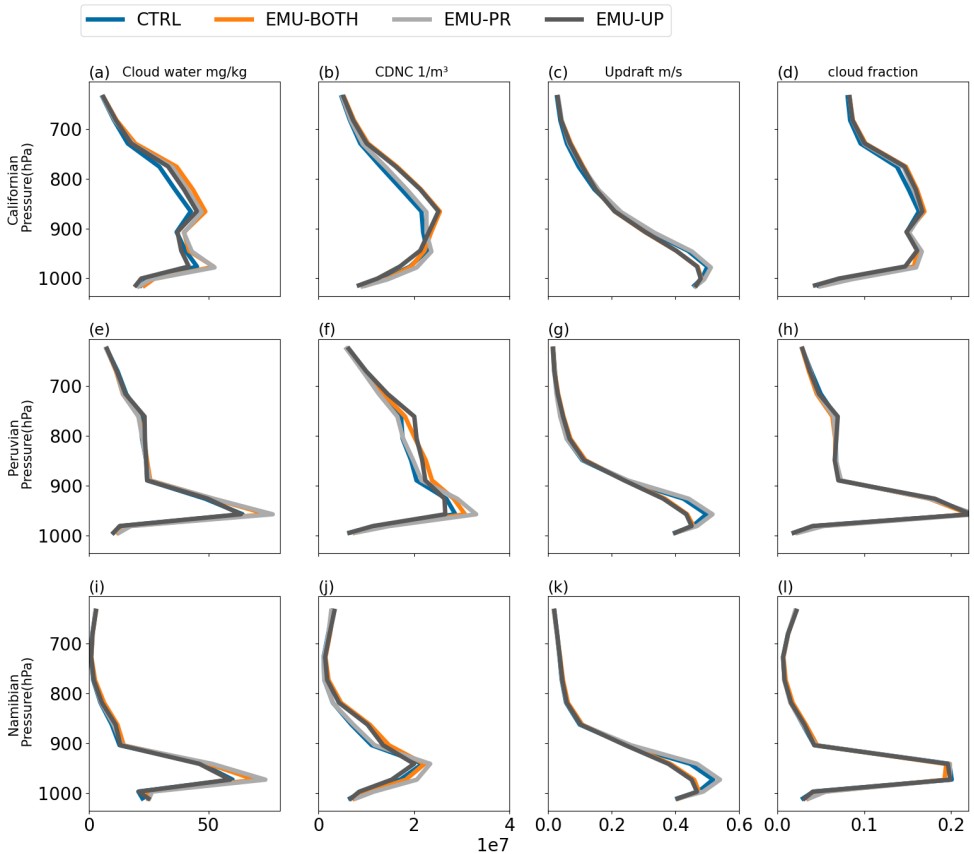

**Figure 6.** Vertical profiles of cloud water (column 1), cloud droplet number concentration (CDNC; column 2), updraft velocity (column 3) and cloud fraction (column 4) for the three stratocumulus regions (rows 1–3). Each line represents a different emulator configuration: CTRL (blue), EMU-BOTH (orange), EMU-PR (gray), and EMU-UP (black).

observations and simulated precipitation for all four experiments. Here, the emulators tend to slightly increase the correlation coefficients and decrease RMSE.

In the three persistent stratocumulus regions (Californian, Peruvian and Namibian), shown in Figs 7b–c and in Table 5, the emulators impact the simulated precipitation quite little in absolute terms. This is expected as precipitation rates from the shallow stratocumulus clouds are typically low as shown in Table 5. The Californian stratocumulus region is an exception because the selected region extends from the dry Californian coast up to the South coast of Alaska where precipitation rates are significantly higher due to the frequent occurrence of mid-latitude low pressure systems (see Fig. 6a). All emulators reduce the clear positive bias in the Namibian stratocumulus region, where the mean surface precipitation rates are the lowest. On the other hand, all emulators increase the negative bias in the Californian stratocumulus region, where the mean surface precipitation

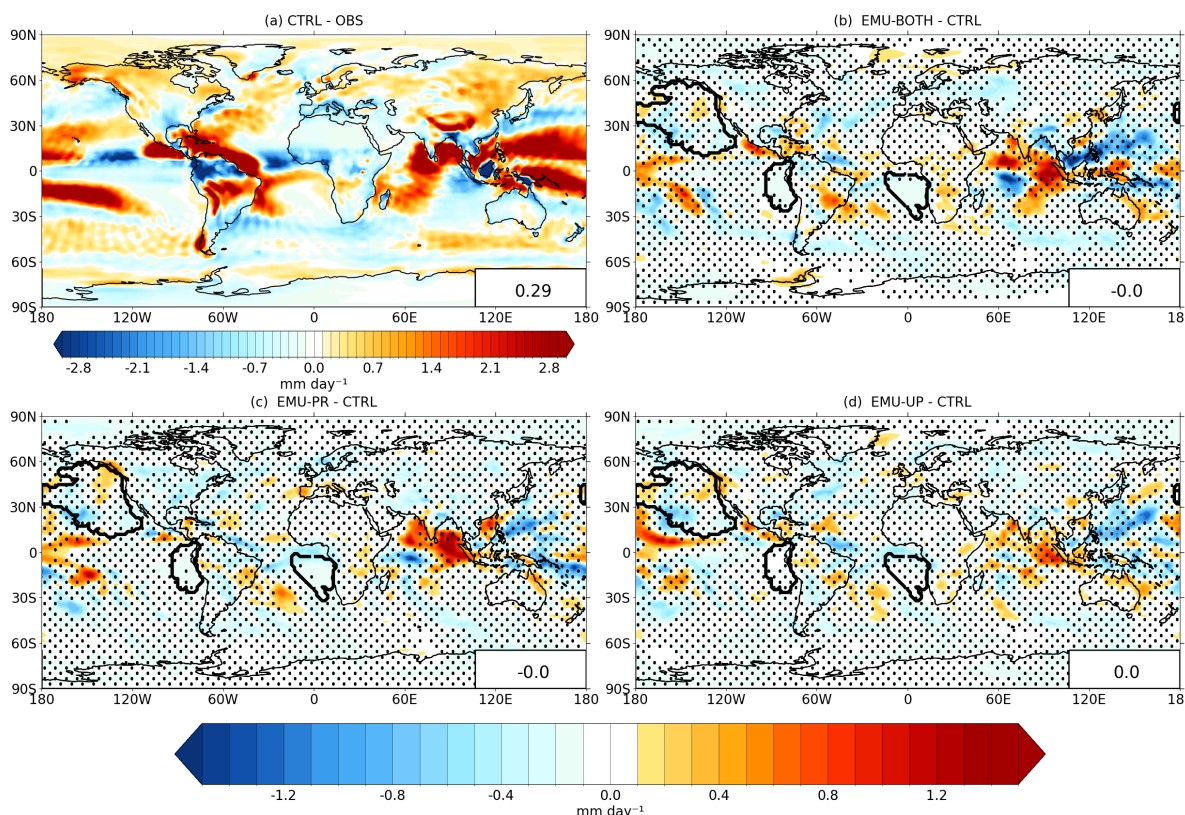

**Figure 7.** Comparison of annual mean surface precipitation ($\mathrm{mm\,day^{-1}}$) between a) the CTRL simulation and observations, b) CTRL and EMU-BOTH, c) CTRL and EMU-PR, and d) CTRL and EMU-UP. Dots indicate areas where the difference is not statistically significant at $p > 0.05$. Identified marine stratocumulus regions are shown in each panel with thick black line. Global mean values are shown in the lower right corners.

rates are the highest. For the Peruvian stratocumulus region, the emulators have both negative and positive impacts on the precipitation bias.

### 3.3.3  Shortwave cloud radiative effects

Figure 8 shows how well the standard ECHAM produces the observed shortwave cloud radiative effect (SW-CRE) and how
420  each applied emulator (EMU-ALL, EMU-PR, and EMU-UP) affects the SW-CRE prediction (note the different color scales). The ECHAM CTRL run captures the observed SW-CRE quite well, although with a slight negative bias in most regions, which results in a global mean bias of -3.71 $\mathrm{Wm^{-2}}$.

**Table 5.** Mean surface precipitation ($\mathrm{mm\,day^{-1}}$) for the three stratocumulus regions and globally from the three different emulator configurations, control run and from observations.

|           | Californian | Peruvian | Namibian | Global |
|-----------|-------------|----------|----------|--------|
| CTRL      | 2.16        | 0.69     | 0.56     | 2.99   |
| EMU-BOTH  | 2.11        | 0.64     | 0.50     | 2.99   |
| EMU-PR    | 2.12        | 0.67     | 0.47     | 2.99   |
| EMU-UP    | 2.11        | 0.70     | 0.51     | 2.99   |
| OBS       | 2.47        | 1.08     | 0.30     | 2.70   |

**Table 6.** Mean SW-CRE values ($\mathrm{W\,m^{-2}}$) for the three stratocumulus regions and globally from the three different emulator configurations, control run and from observations.

|           | Californian | Peruvian | Namibian | Global  |
|-----------|-------------|----------|----------|---------|
| CTRL      | -62.94      | -59.18   | -49.87   | -49.48  |
| EMU-BOTH  | -66.98      | -62.14   | -52.36   | -51.71  |
| EMU-PR    | -65.88      | -62.39   | -52.36   | -50.72  |
| EMU-UP    | -64.00      | -59.49   | -49.72   | -50.54  |
| OBS       | -62.69      | -68.20   | -54.00   | -45.78  |

All emulators increase the negative global mean bias by 1.06–2.23 $\mathrm{W\,m^{-2}}$, with the largest impacts from EMU-BOTH and EMU-PR. The stronger negative SW-CRE in these experiments is primarily caused by the increased LWP (see Table 3). The stronger SW-CRE is mostly seen over mid- and high-latitude marine regions. For example,the Southern Ocean is heavily affected by the precipitation emulator producing a more negative SW-CRE. Table 4 shows correlation coefficients and RMSE between observations and different simulations. The emulators have little effect on the correlations, however, using the emulators increases the RMSE compared to the CTRL simulation.

Although the emulators increase the global mean bias, each emulator has a distinct effect on the three stratocumulus regions as shown in Figs 8b–d. To clarify the differences between the simulations, mean SW-CRE values were calculated for each region, and these are shown in Table 6. Although the global mean bias is negative in CTRL, the bias is positive in the Peruvian and Namibian stratocumulus regions and negative only in the Californian stratocumulus region. Therefore, the increased negative SW-CRE in EMU-PR and EMU-BOTH acts to reduce the bias for the Peruvian and Namibian stratocumulus regions but increases it in the Californian stratocumulus region.

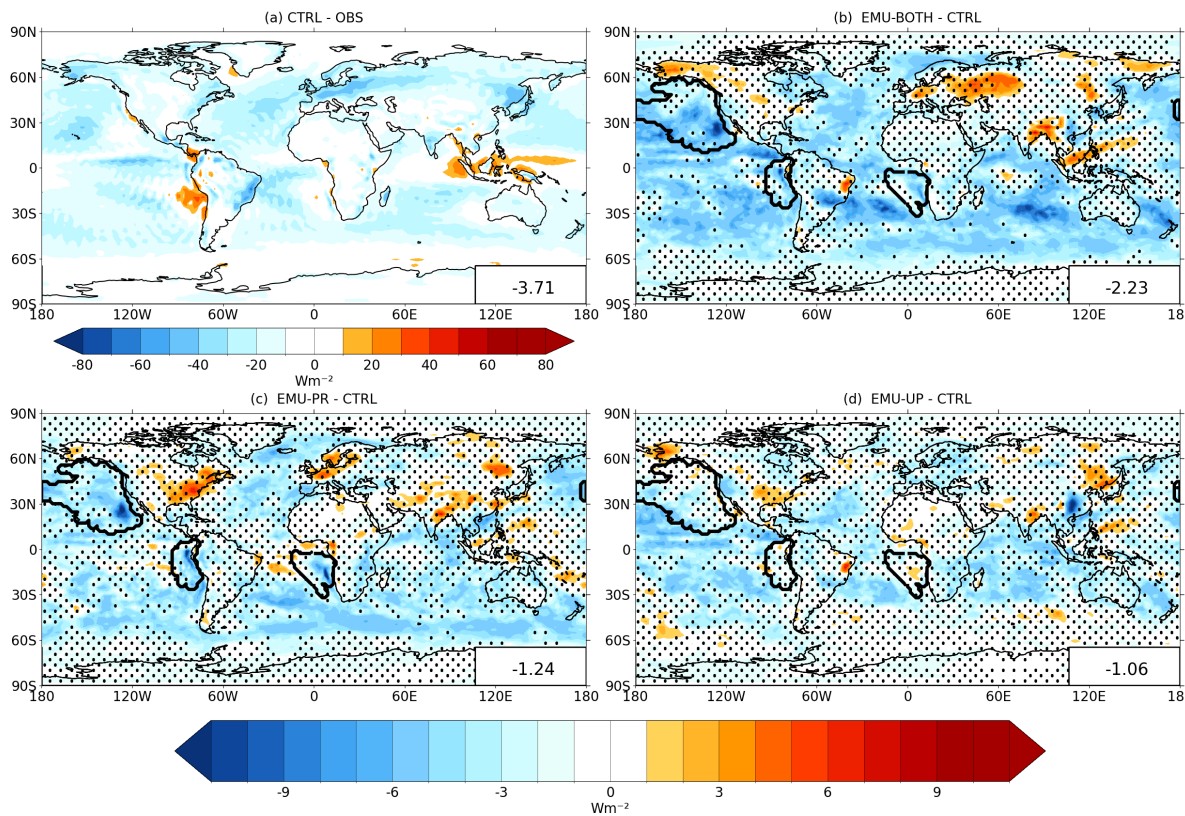

**Figure 8.** Comparison of annual mean SW-CRE between a) CTRL simulation and observations, b) CTRL and EMU-BOTH, c) CTRL and EMU-PR, and d) CTRL and EMU-UP. Dots indicate areas where the difference is not statistically significant at $p > 0.05$. Identified marine stratocumulus regions are shown in panels b–d. Global mean values are shown in the lower right corners.

### 3.3.4 Aerosol effective radiative forcing

Figure 9 shows the calculated aerosol effective radiative forcing (ERF) based on the present-day and pre-industrial EMU-BOTH and CTRL simulations. The global-mean values (-3.04 W m$^{-2}$ for CTRL and -2.94 W m$^{-2}$ for EMU-BOTH) are strongly negative, and their magnitude clearly exceeds the state-of-the art estimates of ERF (e.g., IPCC AR6 WG1 gives a best estimate of -1.3 W m$^{-2}$ with a 90% confidence interval of -2.0 W m$^{-2}$ to -0.6 W m$^{-2}$ over 1750–2014 (Masson-Delmotte et al., 2021)). The very high negative ERF is due to limiting the minimum value of CDNC to 10 cm$^{-3}$ instead of 40 cm$^{-3}$ which is typically used as the lower limit of CDNC in ECHAM. Another factor affecting ERF is the different dependency of autoconversion parameterisation on droplet number concentration employed in ECHAM and emulator training.

Additional details about the aerosol ERF are shown in Table 7, where different components of the ERF are calculated for the three stratocumulus regions and for the whole globe. The impact of the emulators on the aerosol ERF in the stratocumulus

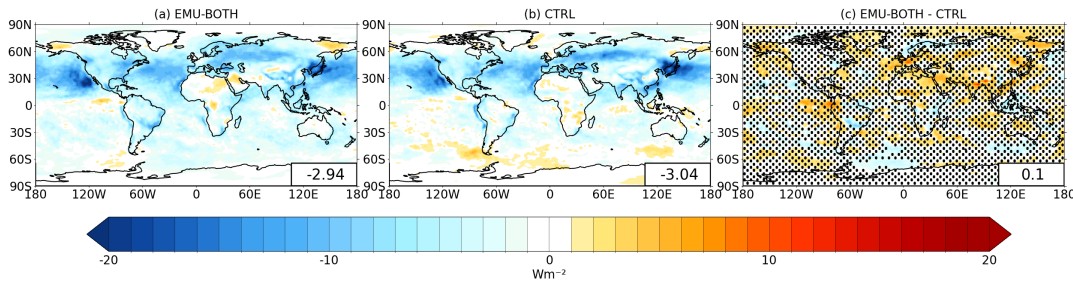

**Figure 9.** Aerosol effective radiative forcing a) when both precipitation and updraft emulators are applied and b) based on the control simulation with the standard ECHAM, and c) the difference between the emulator and control simulations. Global mean values are shown in the lower right corners. Dots in panel c indicate regions where the difference is not statistically significant at $p > 0.05$.

**Table 7.** Mean aerosol effective radiative forcing components ($\text{W m}^{-2}$) for the three stratocumulus regions and globally from the emulator and control runs.

|  | Namibian | | Peruvian | | California | | Global | |
|---|---|---|---|---|---|---|---|---|
|  | EMU-BOTH | CTRL | EMU-BOTH | CTRL | EMU-BOTH | CTRL | EMU-BOTH | CTRL |
| Total | -2.69 | -1.79 | -2.33 | -5.07 | -8.99 | -8.81 | -2.94 | -3.04 |
| Clear | -0.40 | -0.04 | 0.10 | -1.14 | -0.71 | -0.45 | -0.35 | -0.53 |
| Cloud | -2.59 | -1.75 | -2.43 | -3.92 | -8.28 | -8.36 | -2.59 | -2.50 |

445 regions is non-systematic: the ERF becomes more negative for the Namibian and Californian regions but less negative for the Peruvian region.

### 3.3.5 Climate sensitivity

The current emulators were trained for the present-day climate, which raises a question about their performance for warmer climates. To test this, we calculated the impact of the emulators on the climate sensitivity of the ECHAM model. Here, climate

450 sensitivity is estimated using inverse climate change experiments (e.g. Cess et al., 1990), in which a climate change is prescribed by increasing the SST by +4 K and the response in TOA net radiation and near-surface air temperature are evaluated. The climate sensitivity parameter $\lambda$ is then calculated as

$$\lambda = -\frac{\Delta T}{\Delta F_{net}} \tag{2}$$

where $\Delta T$ and $\Delta F_{net}$ are global-mean changes in the near-surface air temperature and TOA net radiation. Here, $\Delta T$ and

455 $\Delta F_{net}$ are evaluated based on the difference between WARM and present-day simulations, separately for the control version

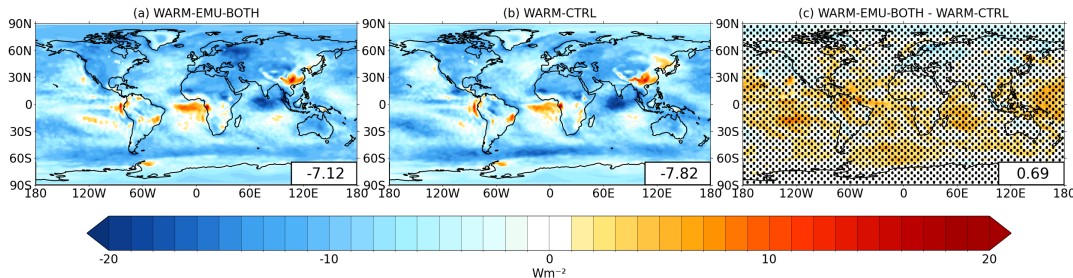

**Figure 10.** The change in TOA net radiation resulting from a +4 K SST change a) when both updraft and precipitation emulators are used, b) based on the control simulation and c) the difference between these two. Dots in panel c indicate regions where the difference is not statistically significant at $p > 0.05$.

of ECHAM (WARM-CTRL vs. CTRL) and for the version that uses both the precipitation and updraft emulators (WARM-EMU-BOTH vs. EMU-BOTH).

Figure 10 shows the change in the TOA net radiation ($\Delta F_{net}$) when emulators are used (panel a) and without emulators (panel b). Although the regional differences (panel c) are not statistically significant, the simulation with emulators tends to produce less negative forcing over the southern hemisphere. The global mean $\Delta T$ for WARM-EMU-BOTH and WARM-CTRL are, 4.47 K and 4.46 K, which results in climate sensitivity parameters of $\lambda$=0.62 K $\left(\mathrm{Wm}^{-2}\right)^{-1}$ and $\lambda$=0.57 K $\left(\mathrm{Wm}^{-2}\right)^{-1}$, respectively. Thus, because of the smaller TOA net flux response, the climate sensitivity is somewhat higher when the emulators are applied.

Figure 11 displays the changes in precipitation and shortwave cloud radiative effect (SW-CRE) in the warm climate simulations. In these simulations, when the climate becomes warmer, changes in precipitation are not significantly influenced by the use of the emulators. The patterns of change in SW-CRE from the control climate to the warmer climate state are also very similar irrespective of whether the emulators are employed or not, but the global-mean change in SW-CRE is more positive when the emulators are used (+2.1 $\mathrm{Wm}^{-2}$) than for the default model version (+1.4 $\mathrm{Wm}^{-2}$).

Overall, while internal variability makes it challenging to robustly identify small changes in model behaviour in 10-year experiments, it can be stated that the emulators have only a modest impact on the modelled climate response. This suggests that these emulators can be used safely for climate states that are, at least, 4 K warmer than the baseline climate from which the training data for the emulators were sampled.

## 4 Discussion: the way forward

In this proof-of-concept study we have shown the applicability of LES-based emulators for precipitation and cloud base updraft velocity predictions in the general circulation model ECHAM. Our simulations are stable and the emulators have a negligible impact on the simulation time. In fact, most of the computational time was spent in developing the emulator, which essentially

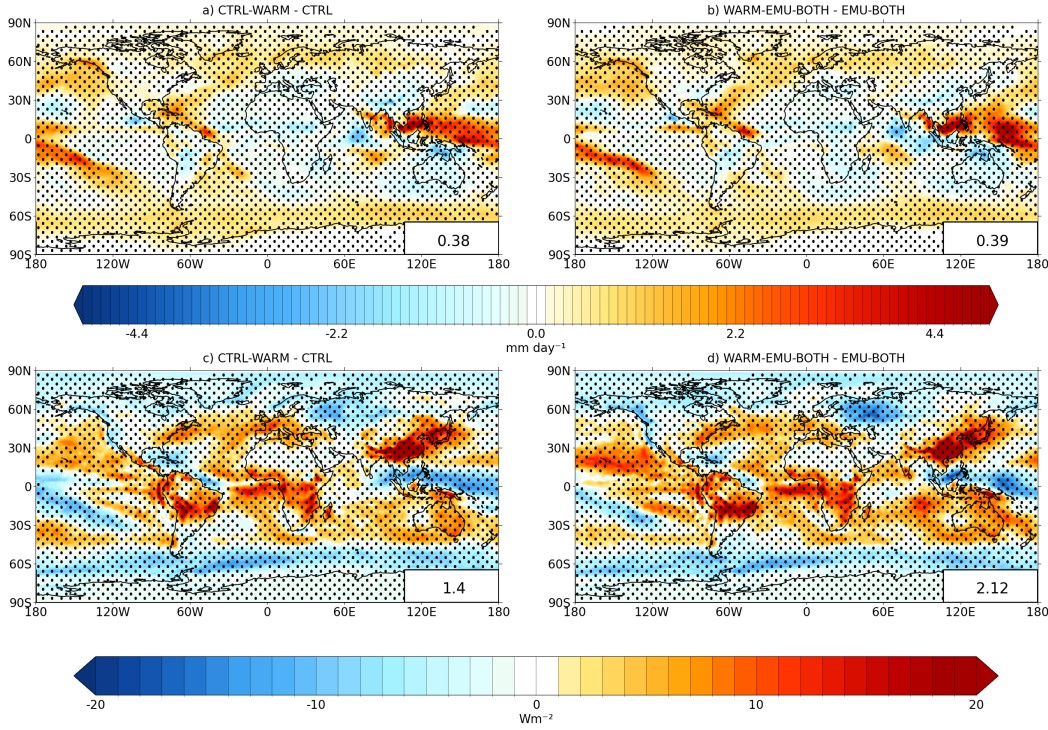

**Figure 11.** Panels a and b show the change in annual mean precipitation for +4 K SST change without emulator (panel a) and with both updraft and precipitation emulator (panel b). Lower panels c and d show the same as above but for SW-CRE. Dots indicate regions where the difference is not statistically significant at $p > 0.05$.

means generating the training data. Applying the emulator to stratocumulus clouds has a relatively small impact on ECHAM simulations, and it is hard to say whether the results are improved or not. The standard ECHAM is tuned so that it matches with observations as well as possible, so even an improved description of processes like precipitation and updraft velocity 480 can reduce the agreement with observations. More quantitative assessment of the potential improvements would have required model re-tuning which is beyond the scope and technical resources of this proof-of-concept study. This would impact results as stratiform rain formation rate by autoconversion, entrainment rate for shallow convection, entrainment rate for deep convection, and convective conversion rate from cloud water to rain are part of the ECHAM6 tuning strategy (Neubauer et al., 2019).

One limitation in developing LES-based emulators for a GCM is that there is a gap in the modelled scales. When a large 485 ensemble of training simulations is needed, the LES especially with detailed microphysics is limited to about 10 km x 10 km domain while most GCMs (including ECHAM) have a grid size in the order of 100 km x 100 km. Most previous studies have used cloud-resolving models (CRMs), which can easily represent a column of a global model. However, CRMs often use similar parameterisations as GCMs, so a LES is needed for real improvements in accounting for turbulence and the details of the aerosol-cloud-precipitation interactions. The high computational cost of using LES with detailed aerosol microphysics

limits its use in the emulator development (Ahola et al., 2022), so in this study we did not account for the aerosol effects. Nevertheless, the approach used here is a promising candidate for the detailed representation of aerosol and cloud related sub-grid processes.

Just like in many other GCMs with computationally expensive aerosol microphysics enabled, coarse vertical resolution is used in the current ECHAM setup. This limits the model's capabilities in reproducing realistic stratocumulus cloud decks (Neubauer et al., 2014), which is an issue for generating the LES input profiles. For that we had to assume a well-mixed cloud-topped boundary layer. Sufficiently high GCM grid resolutions are available only for certain high-resolution model configurations without aerosol microphysics (Chang et al., 2020). Thanks to the increasing computing power, future GCMs can have higher resolution even with aerosol microphysics enabled. In summary, developing LES-based emulators for aerosol-cloud interactions should become easier in the future.

Another issue that is related to the different scales of LES and GCM is the definition of the model parameters. For example, there are different definitions for the mean or characteristic cloud base updraft velocity (Romakkaniemi et al., 2009). Precipitation is even more dependent on model scales and definitions, but at the same time there is great potential for improvements. This is because precipitation is diagnostic in ECHAM, but the LES can include the impacts of sub-grid variability as well as time and aerosol-dependency of rain drop size development as an example. Employing such simulations to train the emulator would add physical realism into large scale models, and reduce the number of commonly employed model tuning parameters needed to reproduce observed cloud cover and properties.

Current emulators produce scalar outputs, such as characteristic updraft velocity, but it would be possible to train additional emulators for distribution parameters like standard deviation or skewness. Indeed, some models (e.g. Golaz et al., 2011) utilize sub-grid scale updraft velocity distributions in calculating CDNC. Due to the nonlinear dependency of activation on vertical velocity, calculations based on a characteristic value and a distribution produce different CDNC.

In broader terms, the main issue in developing a LES-based emulator for a GCM is in finding suitable GCM variables and translating LES outputs into consistent emulator outputs. Surely, not all GCM variables are suitable for emulation, or at least require specific LES setup, and likewise the LES is not the best tool to generate training data for some variables. Even when variables seem consistent, this has to be carefully checked. For example, we had to choose from different updraft velocity definitions the one that matches with the one used in the ECHAM cloud scheme. The LES setup in this case mimics steady-state cloud, where updrafts and rain rate formation rates are fairly constant. Obviously this setup is not valid for parameters like cloud cover, which requires transient simulations. Overall, there are many things that have to be considered case-by-case, but the LES-emulation approach should be a good starting point for most cases.

Finally, there are different machine learning methods that can be used for representing LES simulations. For example, we used Random Forest successfully in our previous study (Ahola et al., 2022). Nevertheless, the binary space partitioning (BSP) sampling combined with the Gaussian process emulation (GPE) technique seems an advantageous combination for our purpose. The BSP method can be used to sample a representative set of the GCM columns based on their likelihood. This means that the emulator is more accurate for those points where the emulator is called most frequently while larger uncertainty

is tolerated for outliers. The GPE technique is suitable for the computationally expensive LES runs requiring, at minimum, only ten simulations per variable.

## 5 Conclusions

Here in this proof-of-concept study we presented the first results of using LES-based precipitation and cloud base updraft velocity emulators in the ECHAM general circulation model simulations. We showed that the emulators, which were applied only to marine stratocumulus clouds, have a small but statistically significant influence on ECHAM simulations. Although the emulation approach has some practical difficulties and limitations, the traditional parameterisations based on simple mathematical functions are not suitable for representing complex dependencies like the output of a LES run. The main advantage of the method is that the LES can account for turbulence and cloud interactions, which are highly parameterised sub-grid scale processes in any large-scale model. The level of details that can be accounted for is largely limited by the LES simulations and their computational costs. In this case, the high computational costs forced us to ignore aerosol-cloud interactions, but the approach is applicable to them as well. In addition, emulators can be trained for presenting other processes than precipitation or cloud-scale updrafts depending on current aims and on the global model. For example, improving mixed-phase cloud physics using LES-based emulators could be the next step ahead.

*Code and data availability.* The LES outputs and ECHAM simulations used in generating the emulator training data can be found from Ahola et al. (2022) and references therein. The codes and the data used in emulator development and validation as well as the ECHAM implementation are available from https://github.com/kallenordling/eclair_emulator (Nordling , 2023). ECHAM simulation results are available from https://a3s.fi/eclair_data_kalle/. ECHAM source code is from https://redmine.hammoz.ethz.ch/projects/hammoz/wiki/Echam630-ham23-moz10. UCLALES-SALSA source code is available from https://doi.org/10.5281/zenodo.5289397 (Tonttila et al., 2021).

*Author contributions.* KN developed the emulators with the help from TR, JPK, MEA, and JA. KN made the ECHAM simulations with help from JPK, HarK, and PR. KN and TR wrote the paper with help from HarK, SR, TR, PR, AL, and HanK. All authors commented on the manuscript. HanK came up with the original concept.

*Competing interests.* The authors declare that they have no conflict of interest.

*Acknowledgements.* This research has been supported by the H2020 European Research Council (ECLAIR (grant no. 646857) and FORCeS (grant no. 821205)) and the Academy of Finland (grant nos. 337552, 322532 and 309127). The authors wish to acknowledge CSC – IT Center for Science, Finland, for computational resources.

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
