# Peer review of "Technical note: Emulation of a large-eddy simulator for stratocumulus clouds in a general circulation model"

_EGUsphere, 2023_

## Referee Comment (RC1)

**Review**

*Technical note: Emulation of a large-eddy simulator for stratocumulus clouds in a general circulation model.* K. Nordling et al.

This paper describes the successful implementation of a process emulator for stratocumulus clouds in a general circulation model (GCM). Details reveal both the ability of the emulator to capture the wide range of stratocumulus in the present-day (PD) climate and the impacts of including it on the climate simulation. Updraft velocities, which are critical for aerosol-cloud interactions and microphysics, are simulated with much greater fidelity to LES by the emulator than by the current ECHAM parameterizations. Using emulators in GCMs is potentially powerful in representing unresolved process important for climate and climate change. Publication with revision is recommended.

**Major Revisions**

1. Although the emulator performs well in the PD climate, with leave-one-out cross validation, an important question remains as to how the emulator would perform in an out-of-sample climate, e.g., globally warmer. Serious problems of this nature have been reported with other emulators, e.g., Rasp et al. (2018, *PNAS*). An experiment in which ECHAM CTRL sea surface temperatures are increased uniformly by 4K, for example, could be run and compared with a corresponding simulation with emulators. Would the CTRL and EMU simulations differ in ways not expected from the PD results? With considerably more effort, the emulators could be trained on both PD and warm climates, and the ability of a model using only PD-trained emulators to reproduce a model using the more broadly trained emulators would provide a sense of the ability of emulators as constructed here to perform well out-of-sample. It may not be feasible to do these experiments in a timely manner for publication of this paper, but the revised version should at least discuss the issues with using the emulators developed here for climate-change experiments.
2. Changes in climate sensitivity between CTRL and EMU are an important issue. In uncoupled models, sensitivity can be assessed using the method of Cess et al. (1990, *J. Geophys. Res.*). This issue should at least be discussed in the revised paper.
3. Fig. 2: Provide correlation coefficient, bias, and RMSE for Emulator vs. LES. Figs. 6 and 7: Provide correlation coefficient and RMSE relative to observations, in addition to bias.

**Minor Revisions**

1. l. 53: Bretherton et al. (2022) train on a global kilometer-scale model, not a super-parameterized model.
2. ll. 101-104: Setting a lower bound of 40 $cm^{-3}$ CDNC to avoid values "considered too low": Is that consideration based on CDNC observations or just a necessity to keep within bounds necessary for realistic simulation of the $20^{th}$ century? If the former, should it be reset to 10?

If the latter, it's worth noting that there remain serious problems with simulating aerosol-cloud interactions requiring a limit not supported by process-level observations and justifying altering it for the emulator experiments.

3. l. 150: The focus in this paper is on a characteristic updraft velocity, as used in some parameterizations for aerosol activation. Activation dependence on vertical velocity is nonlinear, and some parameterizations sample the PDF of updraft velocities. Use of a PDF as opposed to a characteristic value has implications for modeling aerosol-cloud interactions (Golaz et al., 2011, *J. Climate*). Could the emulator approaches described in this paper generate a PDF of updraft velocities from the LES, as opposed to a single characteristic value?

4. l. 155: Clarification of the discussion of the rainfall formation rate would be helpful. The rate appears to be calculated from terms in the rainfall formation budget, instead of from the rainfall itself, and spin-up problems are cited to justify using removal rates. But wouldn't removal rates be problematic also, if autoconversion and accretion rates are? Later in the paper, it becomes evident that the rainfall rate is a major control on modeled clouds, so the question of how the rainfall formation rate is diagnosed from LES results is important.

5. Fig. 4 might convey results more clearly if panel (a) were presented as is, while the other panels showed differences from it.

6. ll. 299-301: How is ECHAM cloud cover obtained from the three-dimensional ECHAM cloud distribution? Are the modeled cloud cover fractions compatible with the observational methods used by Stubenrauch et el. (2013) for comparison?

7. l. 316 and Fig. 5: Between 800 and 850 hPa, CDNC values using EMU-BOTH and EMU-UP differ from CTRL more than the corresponding differences in updraft.

8. l. 423: Question marks appeared on the copy I reviewed where the locations of data from the ECHAM simulations were intended.

Fig. 6 legend: "panels" -> "panel"

l. 330: "extents" -> "extends"

---

## Author Comment (AC1)

We thank the reviewers for their helpful comments for improving our manuscript. The referee comments are shown with *blue font color and italics* , and our point-to-point responses with standard font. Note also that during the review process, we noticed error in the aerosol ERF calculations which are now corrected. Thus the numerical values in section 3.3.4 have changed.

**Anonymous Referee #1**

*This paper describes the successful implementation of a process emulator for stratocumulus clouds in a general circulation model (GCM). Details reveal both the ability of the emulator to capture the wide range of stratocumulus in the present-day (PD) climate and the impacts of including it on the climate simulation. Updraft velocities, which are critical for aerosol-cloud interactions and microphysics, are simulated with much greater fidelity to LES by the emulator than by the current ECHAM parameterizations. Using emulators in GCMs is potentially powerful in representing unresolved process important for climate and climate change. Publication with revision is recommended.*

**Major Revisions**

*1. Although the emulator performs well in the PD climate, with leave-one-out cross validation, an important question remains as to how the emulator would perform in an out-of-sample climate, e.g., globally warmer. Serious problems of this nature have been reported with other emulators, e.g., Rasp et al. (2018, PNAS). An experiment in which ECHAM CTRL sea surface temperatures are increased uniformly by 4K, for example, could be run and compared with a corresponding simulation with emulators. Would the CTRL and EMU simulations differ in ways not expected from the PD results? With considerably more effort, the emulators could be trained on both PD and warm climates, and the ability of a model using only PD-trained emulators to reproduce a model using the more broadly trained emulators would provide a sense of the ability of emulators as constructed here to perform well out-of-sample. It may not be feasible to do these experiments in a timely manner for publication of this paper, but the revised version should at least discuss the issues with using the emulators developed here for climate-change experiments.*

Training emulators on both PD and warm climates would be useful, but as the reviewer says, it is not feasible to develop new emulators for this study. However, we ran the suggested experiment in which ECHAM sea surface temperatures were increased uniformly by 4 K. Those were used to calculate climate sensitivities as suggested below (see the next comment and reply). Climate sensitivities for the ECHAM with and without emulators are similar, which means that the emulators seem to be performing well in these higher temperatures. In addition, we did not see any unexpected behaviour like large changes in precipitation patterns or SW-CRE that could be related to using the emulators at warmer temperatures. The main limitation of these simulations is that the sea ice is fixed, which means that the high latitudes will not become available for the emulators. We added a brief discussion about using emulators in such warmer temperatures.

*2. Changes in climate sensitivity between CTRL and EMU are an important issue. In uncoupled models, sensitivity can be assessed using the method of Cess et al. (1990, J. Geophys. Res.). This issue should at least be discussed in the revised paper.*

We have added a new section about climate sensitivity between CTRL and EMU as suggested by the reviewer. Following the method from Cess et al. (1990, J. Geophys. Res.), we run the ECHAM both with and without emulators in a simulation where sea surface was temperature uniformly increased by 4 K. The difference between climate sensitivities for ECHAM with and without emulators is small.

*3. Fig. 2: Provide correlation coefficient, bias, and RMSE for Emulator vs. LES. Figs. 6 and 7: Provide correlation coefficient and RMSE relative to observations, in addition to bias.*

For rain water production rate (Fig. 2a) Pearson's correlation coefficient, mean error (bias), mean absolute error (MAE) and root mean square error (RMSE) are 0.925, -0.0084 $\mathrm{kg\,m^{-2}\,day^{-1}}$, 0.108 $\mathrm{kg\,m^{-2}\,day^{-1}}$ and 0.526 $\mathrm{kg\,m^{-2}\,day^{-1}}$, respectively. For updraft velocity (Fig. 2b) Pearson's correlation coefficient, bias, MAE and RMSE are 0.914, 0.0011 $\mathrm{m\,s^{-1}}$, 0.033 $\mathrm{m\,s^{-1}}$ and 0.047 $\mathrm{m\,s^{-1}}$, respectively. These are reported in the revised manuscript. We also added a table showing the correlation coefficient and RMSE between observations and all simulations (control and with different emulators) for cloud fraction (new figure), surface precipitation (old Fig. 6) and SW-CRE (old Fig. 7).

**Minor Revisions**

*1. l. 53: Bretherton et al. (2022) train on a global kilometer-scale model, not a super-parameterized model.*

This is now corrected.

*2. ll. 101-104: Setting a lower bound of 40 cm-3 CDNC to avoid values "considered too low": Is that consideration based on CDNC observations or just a necessity to keep within bounds necessary for realistic simulation of the 20th century? If the former, should it be reset to 10? If the latter, it's worth noting that there remain serious problems with simulating aerosol-cloud interactions requiring a limit not supported by process-level observations and justifying altering it for the emulator experiments.*

Simulations were done with the minimum limit for CDNC of 10 $\mathrm{cm^{-3}}$ to allow for more variability in CDNC than with the default value of 40 $\mathrm{cm^{-3}}$. Limiting the lower value of CDNC is commonly used in global models and it is not based on observations (Hoose et al., Geophys. Res. Lett., 2009). There is no strong reason for the choice of 40 $\mathrm{cm^{-3}}$ in the default version of ECHAM; however, this is the value that has been conventionally used in ECHAM since ECHAM4 (Lohmann et al., JGR 1999; this is now cited in the updated manuscript). These authors considered it to be a reasonable lower limit, while acknowledging that lower values can occur over the Southern Ocean. Indeed, the lower limit of 10 $\mathrm{cm^{-3}}$ used in our experiments is more in line with what has actually been observed in extremely clean conditions (see the Introduction in Hoose et al. 2009).

*3. l. 150: The focus in this paper is on a characteristic updraft velocity, as used in some parameterizations for aerosol activation. Activation dependence on vertical velocity is nonlinear, and some parameterizations sample the PDF of updraft velocities. Use of a PDF as opposed to a characteristic value has implications for modeling aerosol-cloud interactions (Golaz et al., 2011, J. Climate). Could the emulator approaches described in this paper generate a PDF of updraft velocities from the LES, as opposed to a single characteristic value?*

Current emulators produce scalar outputs like characteristic updraft velocity (this is what ECHAM uses), but it would be possible to train additional emulators for distribution parameters like standard deviation or skewness. This will be discussed briefly in the revised manuscript.

*4. l. 155: Clarification of the discussion of the rainfall formation rate would be helpful. The rate appears to be calculated from terms in the rainfall formation budget, instead of from the rainfall itself, and spin-up problems are cited to justify using removal rates. But wouldn't removal rates be problematic also, if autoconversion and accretion rates are? Later in the paper, it becomes evident*

*that the rainfall rate is a major control on modeled clouds, so the question of how the rainfall formation rate is diagnosed from LES results is important.*

Accretion is not a problem as it requires rain droplets to form first and the process is limited by the collision rate. This means that droplet growth rates by accretion cannot reach values expected to be unphysically high. However, autoconversion rates depend on cloud droplet size, so in the case of high liquid water content and low CDNC, the rates can be high immediately after the process is switched on after the spin-up. Typically the rates decrease to a reasonable level within 30 min or so, but there are also a few exceptions. Precipitation, on the other hand, depends on the rain droplet size distribution whose development is limited by the accretion process. This means that unrealistic precipitation rates are much less frequent than unrealistic autoconversion rates. Overall, precipitation rate depends on the rain drop size distribution, which is used to calculate the sedimentation velocity distribution. We have clarified the explanation about rain water formation rate in the revised manuscript.

*5. Fig. 4 might convey results more clearly if panel (a) were presented as is, while the other panels showed differences from it.*

We have changed the figure as suggested.

*6. ll. 299-301: How is ECHAM cloud cover obtained from the three-dimensional ECHAM cloud distribution? Are the modeled cloud cover fractions compatible with the observational methods used by Stubenrauch et el. (2013) for comparison?*

ECHAM cloud cover is evaluated using the maximum-random overlap assumption, without applying a satellite simulator. It is acknowledged that this brings some uncertainty to the comparison with satellite data.

*7. l. 316 and Fig. 5: Between 800 and 850 hPa, CDNC values using EMU-BOTH and EMU-UP differ from CTRL more than the corresponding differences in updraft.*

The explanation is related to updraft velocity difference between the emulators and ECHAM. When emulator calls (low clouds) become less frequent with increasing altitude, the impact of emulators become less clear. Because CDNC has strong non-linear dependency on updraft velocity, the impact of the emulators can be seen in CDNC even when all the average updraft velocities start to look alike. This is now briefly discussed in the revised manuscript.

*8. l. 423: Question marks appeared on the copy I reviewed where the locations of data from the ECHAM simulations were intended.*

Code and data availability is now updated.

*Fig. 6 legend: "panels" -> "panel"*
Fixed.

*l. 330: "extents" -> "extends"*
Fixed.

**Anonymous Referee #2**

**General comments**

*The article demonstrates how a computationally light Gaussian Process Emulator is used as a parameterization of shallow clouds in the ECHAM climate model. The emulator was trained on LES simulations. This is important, because the representation of shallow clouds in GCMs is a major source of uncertainty, as they occur on spatial scales smaller than the model grid. The emulator approach is an interesting way of using high-resolution models to inform the parameterization of these clouds in a GCM, and this work demonstrates it in practice. The fact that the emulator can replace a parameterization in a GCM, and run a long time without crashing the model is by itself an achievement.*

*I like the disciplined approach of deciding for which conditions to apply the emulator (low clouds) and using the emulator for only a small number of well-chosen quantities (here vertical velocity and precipitation rate).*

*I have concerns with how the simulations used for emulator training were implemented, detailed below. These should be addressed in the text. I recommend the article is published with a minor revision (and with the simulations as they are).*

**Specific comments**

*The emulator appears designed especially for stratocumulus clouds (title and L58). However the criteria for using the emulator in the GCM are not very strict, and will allow the emulator to be used also for other cloud types, in particular shallow cumulus. I worry that the LESs used for training are small (10x10 km) and run for a short time, 3.5 hours. This means that any mesoscale cloud organization effects (see e.g. Bony et al 2020, https://doi.org/10.1029/2019GL085988) will be missed. For stratocumulus such effects may not be decisive, but for shallow cumulus they are, and then strongly influence the precipitation. Such organization effects are probably not well captured by the original GCM either, so in my view it would be very valuable if the emulator could include them in the parameterization.*

The relaxed conditions for identifying low clouds allow applying the emulators to a reasonably high fraction of ECHAM columns, also beyond archetypal stratocumulus regions. The emulator, however, is effectively limited to stratus and stratocumulus clouds, as the LES simulations were initialized with a cloud fraction of unity. The reason for this is, as the reviewer notes above, the simulation area (10 x 10 km) and the length of the LES simulations (3.5 h) are insufficient to represent properly mesoscale simulations associated with shallow cumulus clouds (e.g., Saffin et al., J. Adv. Mod. Earth Sy., 15, 2023). Producing an emulator that better represents also shallow cumulus clouds would be an important goal for future work, but this is currently beyond our computational resources.

*A second issue related to cumulus is whether the set of input variables to the emulator is adequate also for cumulus. For example, the surface fluxes of heat and moisture might be important parameters for cumulus.*

Please, see the previous reply. The emulator is much better for stratus and stratocumulus clouds than for cumulus.

*The LES simulations are mentioned as the main computational bottleneck in this work. Could you state how expensive they are? My feeling is that longer and larger simulations would be possible with current computational resources, even if 1000s of them are required.*

Each LES simulation took about one hour with 100 CPUs, and running all the 1000 simulations took a couple of weeks (three runs in parallel). Now we could do longer or larger simulations, however, the limit is reached quickly by increasing the domain size or accounting for additional microphysical details.

*Some more details of how the emulator is constructed would be useful. Did you have to or choose to implement it yourself, or was an existing library used? In the source code I saw the GPF library. It's not easy to find the author of it, is there something you could cite?*

We have updated our code and data repositories (please see the updated Code and data availability section) so now it should be easy to find all the details. We also clarify in the manuscript that we used the GPF library (https://github.com/ots22/gpf) extended with our covariance function.

*Does the reference Rasmussen and Williams fully define the algorithms used, or did you make additional design decisions? If so, it would be good to document them, the description is now quite brief.*

Rasmussen and Williams describe both the theory and algorithms. The practical algorithms, the code, and technical details can be found from our updated code and data repositories (links in the Code and data availability section). Additional design decisions not mentioned in the main text include applying standardization to training data and adding a noise term of 1e-9 to stabilise matrix inversions. The optimizer has also parameters related to algorithms, termination tolerance, and parameter bounds.

*Does the emulator give measures of uncertainty? Can such a measure be used to see where additional LES runs would be beneficial, if one wants to extend the training data set?*

Yes, the Gaussian emulator gives uncertainty for its prediction. Where one wants to extend the training data set depends not only on emulator accuracy but also on the frequency of emulator calls. It is desirable that the emulator is more accurate for those points where the emulator is most frequently called while larger uncertainty can be tolerated for outliers. This is what the Binary Space Partitioning (BSP) does. We will clarify this in the revised manuscript.

*Was a new LES dataset constructed for this paper, or were the runs from Ahola 2022 used again? This was mentioned in several places but gave contradictory impressions.*

We use the LES dataset from Ahola et al. (2022). This is now clarified.

*End of section 2.3 - the vertical distribution of precipitation: This seems a good solution to a non-obvious issue that appears in this approach. I appreciate that it is documented here.*

Thanks!

**Minor remarks:**

*L8: "Although especially..." incomplete sentence.*

Changed "properties. Although´´ to "properties, although´´.

*L56: the superparameterization of Jansson et al 2019 (also 2021 https://doi.org/10.1029/2021MS002892) uses an LES specifically aimed at improving the parameterization of shallow clouds (at a high cost and not globally).*

Changed "use LES to improve´´ to ´´use LES and machine learning to improve´´.

*L56: For the discussion of previous approaches and emulators for stratocumulus: Glassmeier et al., 2020 (https://doi.org/10.5194/acp-19-10191-2019) construct a Gaussian Process Emulator for stratocumulus clouds based on LES results not for GCM use but to understand the different states of stratocumulus, and should be mentioned.*

Their work is now mentioned in the introduction.

*L135: the definition of the jumps is hard to understand.*

We clarified (and corrected) that the jumps are differences between maximum and minimum values of total water mixing ratio and liquid water potential temperature near (within the distance of two grid cells) the cloud in an ECHAM column.

*L157: The removal rate procedure is hard to understand.*

We clarified the text regarding the approach, and also changed the term "sedimentation´´ (used in the code) to "precipitation´´, which is a more commonly used term.

*Fig 5: The x axis ticks on the two top rows could be left out, to be consistent with the look of the y-axis and to avoid clutter*

We removed the numerical values on the x axis on the top two rows.

**Code and data availability:**

*I appreciate the code and data being openly available.*

*Line 420: data reference is broken "ECHAM simulation results are available from ? (Nordling et al., ?; last access ?)"*

The data reference has been updated.

*The emulator implementation https://github.com/kallenordling/eclair_emulator): a minimal README and a license statement would be helpful. Additionally, you could consider archiving a specific version in a permanent repository such as Zenodo, which generates a citable DOI. GitHub and Zenodo work well together.*

We have updated the Code and data availability section. We added a README file and a license statement. We have also obtained doi for the GitHub repository by using Zenodo.

*The repository of UCLALES-SALSA could also be mentioned here.*

Reference to the UCLALES-SALSA repository (Tonttila, J, Raatikainen, T., Ahola, J., Kokkola, H., Ruuskanen, A., and Romakkaniemi, S.: UCLALESSALSA/UCLALES-SALSA: Ahola et al., 2021, Zenodo [code], https://doi.org/10.5281/zenodo.5289397, 2021.) has been added.

**Anonymous Referee #3**

*I am genuinely hopeful that new methodologies of the type described in this manuscript will lead to important insights into the physics of the atmosphere and our ability to more effectively model and predict weather and climate. But unfortunately, this particular manuscript does not yet achieve these goals. It can be said that this work often appears as a 'solution looking for a problem'. It may well be that as a proof of concept this technical note may be publishable, but my suggestion is for the authors to consider the following points and improve the manuscript accordingly.*

*1) If the authors wish to reach the parameterization/modeling community (which I hope they do), they should make an effort to improve the description of their approach. These new methods are so different from what is traditionally done by the parameterization and modeling communities that they do require much clearer explanations (including better schematics).*

We have clarified the description of our approach and also improved the schematics (Fig. 1). The updated Code and Data Availability section (and links from therein) contain technical description of the method and also provides our codes and data for the modelling community. We also clarify that emulator development up to LES simulations is already described in our previous publication (Ahola et al., 2022).

*2) A key issue is that there appears to be an inconsistency between the problem/regime that is being addressed and the 'solution' that is being proposed. This link between the 'problem' and the 'solution' needs to be established in a much more effective way - see (3) and (4) below.*

The 'problem' that we aim to address is the lack of subgrid scale physics which is driving shallow stratified clouds in low resolution GCMs. The 'solution' is LES emulation based on data with high enough spatial resolution.

*3) If the focus of the paper is on shallow marine clouds, why would the authors select cloud base updraft velocity (in the context that it is being used) and rain water formation rate as variables for their study? Please clarify.*

We considered cloud base updraft velocity and rain formation for two main reasons. On one hand, these parameters play a major role in the aerosol-cloud interaction (ACI), which is a key uncertainty in the radiative forcing of climate change, and indeed the focus of the projects that provided funding for this work. On the other hand, this choice is pragmatic. These parameters can be reasonably simulated by the LES and the emulated values can be used in ECHAM without major structural changes in the model parameterizations. Other parameters and processes were also considered but they were found more problematic. For instance, cloud fraction would be a key parameter, but emulating it would have required LES setups supporting partial cloud cover, which would have added another level of details. Also, emulating directly the cloud liquid water amount would be challenging, because the emulated values could not be easily used consistently in ECHAM's prognostic cloud scheme.

*4) The marine boundary layer clouds problem in climate and weather models is, to first order, a turbulence-convection-macrophysics problem. So, why are the authors more focused on microphysics?*

As outline above, the focus of this paper was on updraft velocity and precipitation formation, firstly due to their relevance for aerosol-cloud interaction, and secondly, due to the relative simplicity of implementation. A more comprehensive solution to stratocumulus parametrization would indeed require considering the turbulence-convection-macrophysics problem. It would require developing

specific emulators (LES setup and emulator inputs and outputs) for this purpose, and it might also require substantial structural changes in the model PBL, shallow convection and cloud parametrizations. While the scope of our work is more limited than that, it shows that the emulation approach is at least feasible. Such tools can and should be tested with other climate models and model components. Some discussion of this issue will be added.

*5) A demonstration of how far this manuscript is from the more traditional parameterization and modeling research is the sparsity of discussion (and corresponding references) of the work that has been done in this field in the last few decades. This aspect needs to be clearly improved. This work (which is novel and interesting) needs to be grounded in what has been attempted over the last decades and its failures and successes.*

In the introduction we represent the previous work related to climate modelling and machine learning. In the revised manuscript, We have added discussion about the development of traditional cloud parameterizations, with roughly 20 references. We will also give more details about ECHAM and its current parameterizations in section 2.1.2.

*6) LES are extremely powerful tools to help develop and improve parameterizations of turbulence, convection, and cloud macrophysics. But this is not necessarily the case for cloud and aerosol microphysics: LES and other atmospheric models suffer from many similar issues in this context. The LES problems with microphysics have been clearly reported in the literature. So, why are the authors focusing on microphysics? Why should LES be trusted? Please clarify.*

Surely, the double-moment bulk microphysics has its limitations as discussed, for example, by Morrison et al. (J. Adv. Model. Earth. Syst., 2020), but computational costs limit us to use such simple approaches. With increasing computational power we will be able to replace the double-moment microphysics with a more detailed physics also in larger simulation ensembles. Currently the LES with bulk microphysics describes the microphysics of low clouds in much more detail than ECHAM. The current LES (UCLALES) is one of the most commonly used LES in cloud studies, so we have no reason to expect that the model should not be trusted, but obviously there is room for improvements.

*7) Independently of their accuracy, parameterizations and the models in which they are implemented often display internal consistency. The method advocated in this manuscript appears to potentially break this consistency in a variety of ways. Although the authors briefly discuss this issue in the context of the variables that they are focused on, they should openly address this critical issue in broader terms.*

We did consider the problem with internal consistency especially during the emulator development. Naturally, the discussion about internal consistency is focused on those parameters and ECHAM cloud scheme which we used in this study. We added a more general paragraph about consistency. In broader terms, the issue is in translating LES results into GCM variables rather than the emulation method itself. This translation depends on the GCM and the scheme, so there are no generally valid rules. This has to be considered case-by-case, but basically the LES-emulation method that we presented should be a valid starting point for most cases.

*8) The manuscript needs a more explicit discussion of cloud cover and cloud fraction profiles, including adding figures with global maps of cloud cover (such as is done for other variables) and with profiles of cloud fraction (as in figure 5).*

Figures of cloud cover and cloud cover profiles were added, as suggested, along with more discussion on cloud cover. Overall, the maps of cloud cover show that the emulators have a negligible impact

on global cloud cover. Profiles of cloud fraction are shown along with the other profiles (Fig. 5 in the original manuscript). As expected, cloud fraction follows cloud water profiles.